# Excitatory motor neurons are local oscillators for backward locomotion

**Shangbang Gao[1]\*, Sihui Asuka Guan[2,3,4], Anthony D Fouad[5], Jun Meng[2,3,4], Taizo Kawano[2†], Yung-Chi Huang[6], Yi Li[1], Salvador Alcaire[2,3,4], Wesley Hung[2], Yangning Lu[2,3,4], Yingchuan Billy Qi[7], Yishi Jin[7], Mark Alkema[6], Christopher Fang-Yen[5,8], Mei Zhen[2,3,4]\***

[1]Key Laboratory of Molecular Biophysics of the Ministry of Education, College of Life Science and Technology, Huazhong University of Science and Technology, Wuhan, China; [2]Lunenfeld-Tanenbaum Research Institute, Mount Sinai Hospital, Toronto, Canada; [3]Department of Molecular Genetics, University of Toronto, Toronto, Canada; [4]Department of Physiology, University of Toronto, Toronto, Canada; [5]Department of Bioengineering, School of Engineering and Applied Science, University of Pennsylvania, Philadelphia, United States; [6]Department of Neurobiology, University of Massachusetts Medical School, Worcester, United States; [7]Neurobiology Section, Division of Biological Sciences, University of California, San Diego, United States; [8]Department of Neuroscience, University of Pennsylvania, Philadelphia, United States

**\*For correspondence:**
sgao@hust.edu.cn (SG);
zhen@lunenfeld.ca (MZ)

**Present address:** [†]Graduate School of Science, Kobe University, Kobe, Japan

**Competing interests:** The authors declare that no competing interests exist.

**Abstract** Cell- or network-driven oscillators underlie motor rhythmicity. The identity of *C. elegans* oscillators remains unknown. Through cell ablation, electrophysiology, and calcium imaging, we show: (1) forward and backward locomotion is driven by different oscillators; (2) the cholinergic and excitatory A-class motor neurons exhibit intrinsic and oscillatory activity that is sufficient to drive backward locomotion in the absence of premotor interneurons; (3) the UNC-2 P/Q/N high-voltage-activated calcium current underlies A motor neuron's oscillation; (4) descending premotor interneurons AVA, via an evolutionarily conserved, mixed gap junction and chemical synapse configuration, exert state-dependent inhibition and potentiation of A motor neuron's intrinsic activity to regulate backward locomotion. Thus, motor neurons themselves derive rhythms, which are dually regulated by the descending interneurons to control the reversal motor state. These and previous findings exemplify compression: essential circuit properties are conserved but executed by fewer numbers and layers of neurons in a small locomotor network.
DOI: https://doi.org/10.7554/eLife.29915.001

## Introduction

Central pattern generators (CPGs) are rhythm-generating neurons and neural circuits with intrinsic oscillatory activities. Across the animal phyla, CPGs underlie rhythm of motor behaviors that are either continuous, such as breathing and heartbeat, or episodic, such as chewing and locomotion (*Grillner, 2006*; *Grillner and Wallén, 1985*; *Kiehn, 2016*; *Marder and Bucher, 2001*; *Marder and Calabrese, 1996*; *Selverston and Moulins, 1985*). CPGs that drive locomotor behaviors generally require signals from the central or peripheral nervous systems for initiation or reconfiguration (*Pearson, 1993*).

The concept of locomotor CPGs originated from the observation that decerebrate cats could sustain rhythmic hind limb muscle contraction (*Brown, 1911*, *Brown, 1914*). In deafferented locusts, flight motor neurons (MNs) exhibited rhythmic activity in response to non-rhythmic electrical

stimulation (*Wilson, 1961*). Isolated nerve or spinal cords from the leech (*Briggman and Kristan, 2006*), lamprey (*Wallén and Williams, 1984*), rat (*Juvin et al., 2007*; *Kiehn et al., 1992*) and cat (*Guertin et al., 1995*) are able to generate rhythmic MN activity and/or fictive locomotion. These findings imply that locomotor systems can intrinsically sustain rhythmic and patterned electrical activitives, independent of inputs from the descending neural networks or sensory organs.

Locomotor CPGs have been located in several animals. For most invertebrates, they consist of premotor interneurons (INs) that drive MN activity (*Marder and Bucher, 2001*). In vertebrates, they reside in the spinal cords, where multiple pools of premotor INs instruct and coordinate the output of different MN groups (*Grillner, 2006*; *Kiehn, 2006, 2016*). MNs can retrogradely regulate the activity of CPG INs or premotor INs in the crayfish (*Heitler, 1978*), leech (*Rela and Szczupak, 2003*; *Szczupak, 2014*), *C. elegans* (*Liu et al., 2017*) and zebrafish (*Song et al., 2016*) in a conserved, mixed electric and chemical synapse configuration.

*C. elegans* generates rhythmic and propagating body bends that propel either forward or backward locomotion. Synaptic wiring of the adult locomotor system has been depicted by serial electron microscopy (*White et al., 1976*, *White et al., 1986*). There are five MN classes: A, B, D, AS and VC in the ventral nerve cord. The A (A-MN), B (B-MN), and D (D-MN) classes contribute the vast majority of neuromuscular junctions (NMJs) in the body. Each class is divided into subgroups that innervate dorsal or ventral muscles. Repeated motor units comprised of the A-, B- and D-MNs make tiled dorsal and ventral NMJs along the body.

Both B- and A-MNs are cholinergic and excitatory, potentiating muscle contraction (*Gao and Zhen, 2011*; *Liu et al., 2011*; *Nagel et al., 2005*; *Richmond and Jorgensen, 1999*; *White et al., 1986*), whereas D-MNs are GABAergic and inhibitory, promoting muscle relaxation (*Gao and Zhen, 2011*; *Liewald et al., 2008*; *Liu et al., 2011*; *McIntire et al., 1993*). B- and A-MNs form dyadic synapses, with body wall muscles and D-MNs that in turn contralateral muscles as co-recipients. This configuration supports reciprocal dorsal-ventral cross-inhibition for body bending (*White et al., 1986*).

Descending and ascending premotor INs innervate excitatory MNs. Three pairs of INs - AVA, AVB, and PVC - extend axons along the ventral nerve cord, and form synapses to all members of the MN classes that they partner with. They contribute to two sub-circuits: a forward movement-promoting unit, where AVB and PVC make electrical and chemical synapses with the B-MNs, respectively, and a backward movement-promoting unit, where AVA innervate the A-MNs through both electrical and chemical synapses (*Chalfie et al., 1985*; *White et al., 1986*; illustrated in *Figure 1A*). Reciprocal inhibition between the two sub-circuits underlies stabilization of, and transition between the forward and reversal motor states (*Kato et al., 2015*; *Kawano et al., 2011*; *Roberts et al., 2016*).

However, a fundamental question remains unanswered: the origin of motor rhythm. Despite an extensive understanding of *C. elegans* anatomy and physiology (*Duerr et al., 2008*; *McIntire et al., 1993*; *Pereira et al., 2015*; *Rand, 2007*; *White et al., 1986*), the existence and identity of its locomotor CPGs remain speculative. CPGs were proposed to reside in the head, based on an observation that flexion angles decay toward the tail during both forward and backward locomotion (*Karbowski et al., 2008*). Several INs affect bending (*Bhattacharya et al., 2014*; *Donnelly et al., 2013*; *Hu et al., 2011*; *Li et al., 2006*), but none are essential for activation or rhythmicity of locomotion. The remarkable biomechanical adaptability of *C. elegans* locomotion (*Fang-Yen et al., 2010*) implied a prominent role of proprioceptive feedback, irrespective of the presence or location of oscillators (*Gjorgjieva et al., 2014*). Indeed, B-MNs were modeled to be activated solely by proprioception to propagate body bends during forward locomotion (*Cohen and Sanders, 2014*; *Wen et al., 2012*). However, there has been no direct experimental evidence on whether *C. elegans* oscillators exist, where they are encoded, and how they contribute to motor rhythm (reviewed in *Zhen and Samuel, 2015*).

Here, we addressed these questions through cell ablation, electrophysiology, genetics, and calcium imaging. We reveal that *C. elegans* has multiple oscillators, and different oscillators drive forward or backward locomotion. For backward locomotion, the A-MNs themselves constitute a distributed network of oscillators. Inhibition and potentiation of their intrinsic activities, either through manipulating UNC-2, the *C. elegans* P/Q/N-high-voltage-activated calcium channel (VGCC)

that underlies their oscillation, or, through manipulating the synaptic inputs from their descending premotor INs AVA, alter the velocity, propensity and maintenance of the reversal motor state. We conclude that the *C. elegans* motor circuit is a CPG-driven locomotor network, except that its rhythm is derived from excitatory MNs.

In most locomotor networks, the CPG premotor INs and MNs exhibit rhythmic action potential bursts that correlate with fictive or non-fictive locomotion (*Grillner, 2006*; *Kiehn, 2016*), but all examined *C. elegans* neurons to date do not fire classic action potentials (*Bargmann, 1998*; *C. elegans Sequencing Consortium, 1998*; *Goodman et al., 1998*; *Kato et al., 2015*; *Liu et al., 2017*; *Xie et al., 2013*). Results from this study, together with findings that *C. elegans* body wall muscles generate calcium-dependent action potential bursts (*Gao et al., 2015*; *Gao and Zhen, 2011*; *Liu et al., 2013*), unveil a simplified, but fundamentally conserved underpinning of locomotion: the ventral cord MNs assume the role of rhythm generators, instructing action potential bursting in the body wall muscles (see Discussion).

Findings from these studies, and other systems (see Discussion) support the notion that compression - a single neuron or neuron class assuming the role of multiple layers in complex neural circuits - is a general property of small circuits. This property allows small nervous systems to serve as compact models to dissect the organizational logic of neural circuits, exemplified by the intricate roles of a mixed synapse configuration conserved not only at the locomotory, but also other motor circuits (see Discussion).

## Results

### Motor neurons sustain body bends in the absence of premotor INs

To address whether and where *C. elegans* oscillators are present, we first examined the behavioral consequence of ablating MNs or premotor INs. In previous studies, ablation was restricted to a few neurons and in a small number of animals (*Chalfie et al., 1985*; *Kawano et al., 2011*; *Rakowski et al., 2013*; *Roberts et al., 2016*; *Wicks and Rankin, 1995*; *Zheng et al., 1999*). With a flavoprotein miniSOG, which induces acute functional loss, subsequent death and anatomic disappearance of neurons by photo-activation (*Qi et al., 2012*), we ablated the entire population of premotor INs or the A/B/D-MNs (Materials and methods).

We describe their impact on locomotion by two parameters: velocity, defined as the mid-point displacement over time; curvature, defined as the bending angles of body segments (Materials and methods; *Figure 1A–D*). Without premotor INs, animals lost motility, as represented by reduced mid-point displacement (*Figure 1Dii*; *Figure 1—figure supplement 1Aiv; Video 1*). These animals, however, were not paralyzed. Their posture recapitulated that of a class of mutants called *kinkers* (*Kawano et al., 2011*): the head oscillates, but oscillation does not propagate along the body; the body bends, but bending is disorganized (*Figure 1Aii–Cii*; *Figure 1—figure supplement 1Ai-iii*). In *kinker* animals, the stalled mid-point displacement is not caused by a lack of motor activity, but instead by the competiton between the antagonistic forward- and reversal-driving motor neuron activity (*Kawano et al., 2011*). The *kinker*-like posture indicates that animals without premotor INs retain locomotor activity (*Figure 1Dii*; *Figure 1—figure supplement 1Aiv,B*). Indeed, calcium imaging confirmed that these animals exhibited robust activity in body wall muscles (*Video 1*).

Upon the removal of all A/B/D-MNs, animals also lost motility (*Figure 1Diii*, *Figure 1—figure supplement 1B*). Similar to animals without premotor INs, their head oscillation persisted. Unlike the premotor IN-less animals, however, their body bending was attenuated, resulting in an oscillating head pulling forward a non-bending body (*Figure 1Biii*). Attenuation of body bends concurs with the anatomy - the A/B/D-MNs together contribute the majority of NMJs to body wall muscles.

The persistence of head oscillations upon ablation of either all premotor INs or most ventral cord MNs implicates a separation of oscillators for the head and body. CPGs for head oscillation may promote forward locomotion (*Karbowski et al., 2008*; *Pirri et al., 2009*). The persistence of body bends in premotor INs-less animals suggests that some ventral cord MNs may be intrinsically active.

## A-MNs generate rhythmic backward locomotion in the absence of premotor INs

To identify the MN groups with autonomous activity, we next ablated premotor INs in conjunction with the A-, B-, or D-MNs, respectively, to compare their effects on locomotion.

Ablation of individual MN classes in premotor IN-less animals resulted in drastically different motor outputs. Upon ablation of premotor INs and A-MNs, animals exhibited predominantly sluggish forward locomotion: an oscillating head slowly pulled a body with shallow bending (*Figure 2Aii, Bii*; *Video 2* part 3–4). When premotor INs and B-MNs were removed, animals exhibited predominantly backward locomotion with robust body bending (*Figure 2Ai,Bi*; *Video 2* part 1–2). Their reversals were periodically stalled by the forward-promoting head oscillations that interfered with the mid-point displacement (*Video 2* part 1–2). Removing D-MNs in premotor IN-less animals did not reduce bending – they maintained a *kinker*-like posture without directional movements (*Figure 2Aiii,Biii*; *Video 2* part 5–6).

These results indicate that, in the absence of all premotor INs, body bends primarily originate from the A-MNs (see Discussion). Strikingly, A-MNs drive backward locomotion with robust bending in the absence of premotor INs (*Figure 2*, *Figure 1—figure supplement 1C*), implying that their endogenous activities suffice for both execution of body bends, and organization of bending propagation.

## Sparse removal of A-MNs alters, but does not abolish, backward locomotion

Because A-MNs can organize backward movements without premotor INs, they may function as either a chain of phase-locked oscillators, or as flexors, where each neuron is sequentially activated by bending of its neighboring body segment. These possibilities can be distinguished by examining the effect of sparse removal of A-MNs. In the former case, the removal of individual A-MNs should alter, but not prevent, bending propagation during backward locomotion. In the latter case, bending propagation shall stall at most posterior body segments that the ablated A-MNs innervate.

Our ablation results (*Figure 3*; *Figure 3—figure supplement 1*; *Video 3*) favor the oscillator hypothesis. Removal of A-MNs in anterior body segments did not prevent the initiation and propagation of reversal waves in the mid- and posterior body segments (*Figure 3C–1i, 1ii*; *Figure 3—figure supplement 1B–1iii*). The head and tail exhibited independent reversal bending waves upon the ablation of mid-body A-MNs. When most or all mid-body A-MNs were ablated, the head exhibited either high (*Figure 3C–2i, 2ii*; *Figure 3—figure supplement 1B–1i*) or low (*Figure 3—figure supplement 1B–2i*) frequency oscillations that were uncoupled in phase with slow tail-led bending. Ablation of a few mid-body MNs also led to uncoupling between the anterior and posterior body bends, but many tail-initiated bending waves propagated through to the head (*Figure 3—figure supplement 1B–2iii*). When posterior A-MNs were removed, bending initiated and propagated from body segments anterior to ablated areas (*Figure 3C–3i, 3ii*; *Figure 3—figure supplement 1B–1ii, 2ii*).

Quantitatively, ablation of A-MNs from either the anterior, mid-body, or posterior segments (*Figure 3A*; *Figure 3—figure supplement 1A*)

Premotor IN MiniSOG (-LED)
(*hpIs321;hpIs331*)

Control

From Figure 1B–1

Part 1 of 3

**Video 1.** Locomotor behaviors (Part 1 and Part 2) and calcium imaging of the body wall muscles (Part 3) of *C. elegans* without premotor INs. (Part 1, 2) Upon ablation of all premotor INs, animals exhibit kinked posture and uncoordinated local body bends; head oscillations persist but fail to propagate along the body. (Part 3) Calcium imaging of body wall muscles was carried out in transgenic animals after the ablation of all premotor INs. Calcium activity persists in muscles, and high calcium activity corroborates with body bending. Left panel: RFP signals in body wall muscles (with extra signals in the gut from the miniSOG transgene). Right panel: GCaMP3 signals in body wall muscles.
DOI: https://doi.org/10.7554/eLife.29915.004

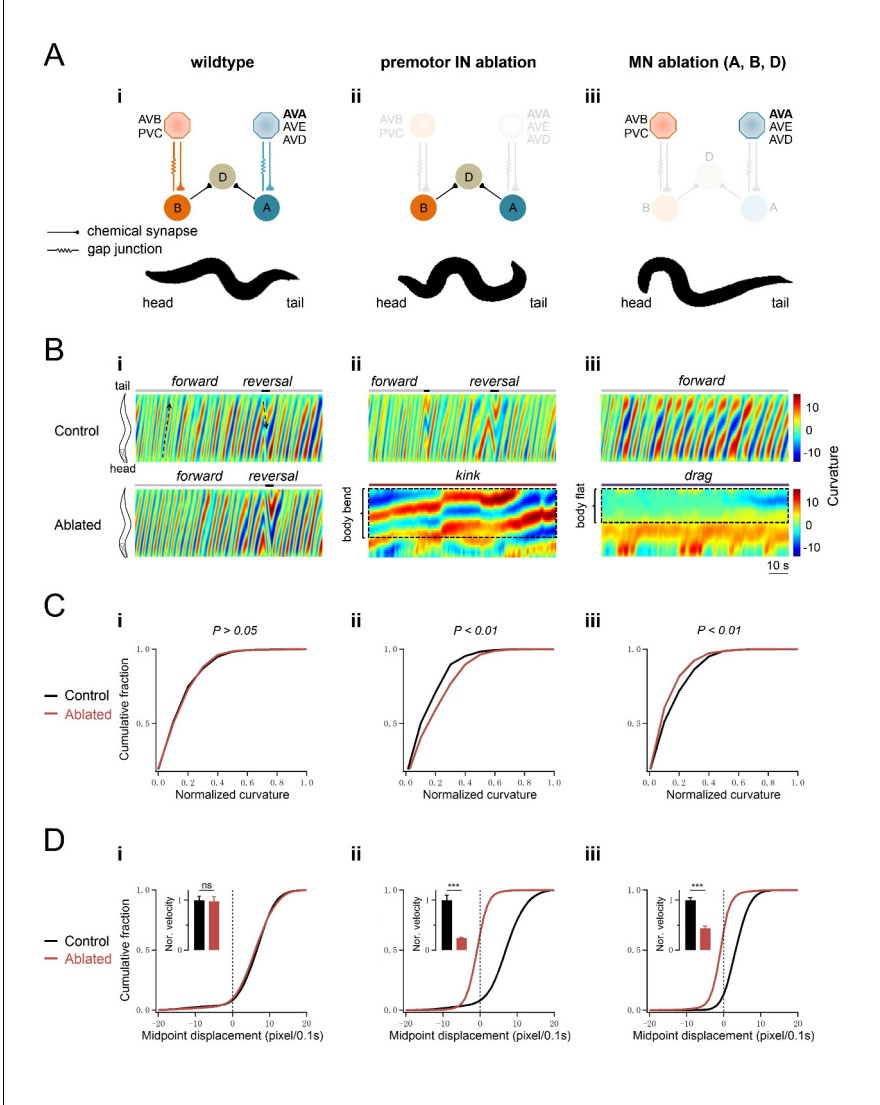

**Figure 1.** Body bends persist upon the ablation of premotor INs. (**A**) The removal of premotor INs or MNs exerts different effects on body bends. *Upper panel*: schematics of the *C. elegans* motor circuit components and connectivity in wildtype animals (i) and upon ablation of respective neuronal populations (ii, iii). Hexagons and circles represent premotor INs and ventral cord MNs, respectively. Orange and blue denote components of the forward and reversal motor circuit, respectively. Taupe denotes neurons that participate both forward and backward locomotion. *Lower panel*: a snap shot of representative body posture exhibited by adult *C. elegans* with intact motor circuit (i), and upon the ablation of premotor INs (ii) or MNs (iii). (**B**) Representative curvature kymograms along the body of moving animals in respective genetic backgrounds. The upper and lower panels denote animals without (Control) and with (Ablated) LED illumination-induced neuron ablation during development. Black arrows on kymograms denote the direction of body bend propagation. (i) Wildtype (N2) animals exhibit a preference for continuous forward locomotion, consisting of anterior to posterior body bend propagation, with occasional and short backward locomotion, exhibited as posterior to anterior body bend propagation; (ii) ablation of all premotor INs (Ablated) leads to stalled body bending that prevents the propagation of head bending; (iii) simultaneous ablation of three major MN classes largely eliminates body bending in regions posterior to head. (**C**) Distribution of body curvatures posterior to head (33–96% anterior-posterior of body length) in wildtype (i), premotor INs-ablated (ii), and MNs-ablated (iii) animals, with (Control) and without (Ablated) LED illumination-induced neuronal ablation. Premotor INs ablation leads to an increase (ii) whereas MN ablation a decrease (iii) of the bending curvature. (**D**) Distribution of instantaneous velocity, represented by centroid displacement, in wildtype (i) premotor INs-ablated (ii) and MNs-ablated (iii) animals, with (Control) and without (Ablated) neuronal ablation. The ablation of either premotor INs or MNs leads to a drastic

*Figure 1 continued on next page*

*Figure 1 continued*
reduction of velocity. *n* = 10 animals per group (**C, D**). p>0.05 (not significant), ***p<0.001 against the respective non-ablated Control group by the Kolmogorov-Smirnov test.
DOI: https://doi.org/10.7554/eLife.29915.002
The following figure supplement is available for figure 1:

**Figure supplement 1.** (A) Locomotor phenotypes of animals upon the ablation of premotor INs and (**B,C**) the propensity of directional movements in animals of respective genotypes.
DOI: https://doi.org/10.7554/eLife.29915.003

caused significantly decreased local reversal wave speed, but modest or negligible change in wave speed in other body segments (*Figure 3D*). Thus, rhythmic body bends can arise from multiple locations, further supporting the presence of a chain of reversal oscillators.

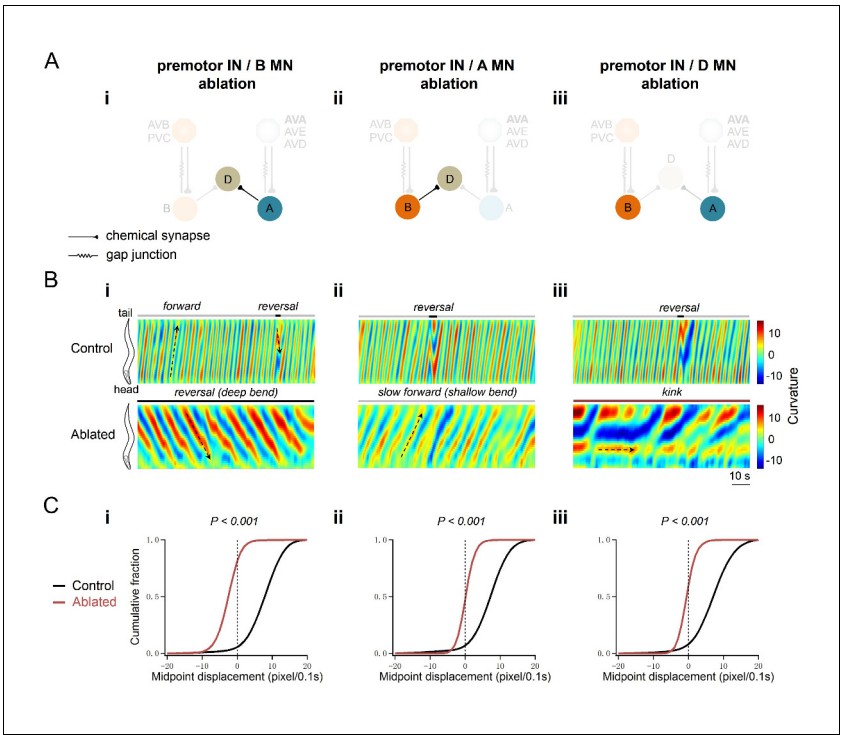

**Figure 2.** A-MNs execute directional, rhythmic locomotion without premotor INs. (**A**) Schematics of the motor circuit components and connectivity of animals of respective genotypes, upon co-ablation of premotor INs and B-MNs (i), premotor INs and A-MNs (ii), or premotor INs and D-MNs (iii). (**B**) Representative curvature kymograms along the entire length of moving animals, without (Control) and with (Ablated) LED illumination-induced neuronal ablation. Black arrows pointing upwards and downwards on kymograms denote posteriorly or anteriorly propagating body bending, respectively. The horizontal arrow denotes the absence of bending propagation. Animals without premotor INs and B-MNs (i, lower panel) exhibit backward locomotion, as posterior to anterior propagating deep body bends, regardless of the propagation direction of head bending. Those without premotor INs and A-MNs (ii, lower panel) often exhibit slow forward locomotion, consisted of slowly propagating, anterior to posterior, shallow body bends. Animals without premotor INs and D-MNs (iii, lower panel) lead to *kinker* postures as in premotor INs-ablated animals. (**C**) Distribution of instantaneous velocity of animals of respective genotypes, represented by the mid-point displacement. The forward velocity (ii) is drastically reduced, whereas reversal velocity (i) is less affected than upon abaltion of premotor INs. *n* = 10 in each group, p<0.001 against the respective non-ablated Control group by the Kolmogorov-Smirnov test.
DOI: https://doi.org/10.7554/eLife.29915.005

## A-MNs exhibit oscillatory activities independent of premotor IN inputs

CPGs should exhibit self-sustained oscillatory activity. We sought experimental evidence for such a property by performing electrophysiology and calcium imaging analyses on A-MNs.

First, we examined a dissected neuromuscular preparation consisting of an exposed ventral nerve cord and the body wall muscles that they innervate (*Gao et al., 2015*; *Richmond and Jorgensen, 1999*). The majority of excitatory NMJs are made by the A- and B-MNs in this preparation. Prior to ablation of premotor INs and B-MNs, whole-cell voltage clamp of the ventral body wall muscles revealed ~25 Hz, ~ -10 pA miniature postsynaptic currents (mPSCs) (*Figure 4—figure supplement 1A,B*). Upon ablation, the mPSC frequency exhibited a moderate ~30% reduction (*Figure 4—figure supplement 1A,B*). More important, however, was that in 70% of preparations (*n* = 10), we observed periodic rhythmic PSC (rPSC) bursts (*Gao et al., 2015*) at ~90 s intervals (*Figure 4A–C*). The rPSC bursts were distinct from high frequency mPSCs: each lasted 2–3 s, consisting of 3–10 Hz, -50 - -300 pA depolarizing currents (*Figure 4A–C*; Figure 6A). By contrast, only in 10% non-ablated preparations (*n* = 10), we observed PSC bursts of similar characteristics, but as sporadic single-unit events. These results suggest that A-MNs generate periodic electrical activities without premotor INs, and premotor INs may inhibit A-MN rhymthic activity.

In the current clamp configuration, we observed periodic action potential bursts that corresponded to the rPSC bursts after premotor INs and B-MNs were ablated (*Figure 4—figure supplement 1C–E*). Muscle action potentials correlate with contraction (*Gao and Zhen, 2011*), confirming the physiological relevance of the autonomous MN activity to locomotion. The notion that most periodic rPSC bursts originate in A-MNs was confirmed by comparing with preparations in which premotor INs and A-MNs were ablated: they exhibited 60% reduction in mPSC frequency, but only sporadic PSC bursts were detected in 17% preparations (*n* = 11).

CPGs exhibit oscillatory activity in the presence of sustained inputs. We further examined the effect of direct stimulation of excitatory MNs by a light-activated rhodopsin Chrimson (*Klapoetke et al., 2014*). In these preparations, sustained activation of the ventral muscle-innervating both A- and B-MNs at high light intensity led to high-frequency PSCs, reflecting potentiated synaptic vesicle release (*Figure 4—figure supplement 2*). Upon sequential reduction of the light intensities, however, the stimulation of A-MNs, but not of B-MNs, began to exhibit rPSC bursts (*Figure 4—figure supplement 2A*).

Results from A-MN calcium imaging in intact animals also support their oscillatory property. To reduce potential effects of proprioceptive coupling, we recorded the A-MN activity from intact animals that were fully immobilized using the surgical glue. While sporadic calcium activities were occasionally observed for some A-MNs in non-ablated wildtype control animals, robust calcium oscillation was revealed in A-MNs in all animals after the co-ablation of premotor INs and B-MNs (*Figure 4D–F*). Individual A-MNs exhibited large variations in amplitudes of calcium oscillation (Figure R2; see Response letters), but shared the ~50 s oscillatory cycle (*Figure 4F–H*). DA9, which innervates the most posterior dorsal muscles (*Figure 4D*; *Video 4* ), exhibited the highest calcium activity.

Lastly, we performed simultaneous calcium imaging of a cluster of posteriorly positioned A-MNs in moving animals, after we removed all premotor INs and B-MNs (*Figure 5A*). Consistent with the notion that A-MNs' endogenous

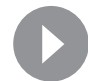

Premotor and B motor neuron miniSOG (-LED)

(*hpIs321;hpIs331;hpIs372*)

Control

From Figure 2B-I

Part 1 of 6

**Video 2.** Co-ablation of premotor INs with either A-, B- or D-MNs leads to different locomotor behaviors. (Parts 1, 2) Upon co-ablation of the premotor INs and B-MNs, animals exhibit sluggish forward locomotion where the body passively follows head oscillation. (Parts 3, 4) Upon the co-ablation of premotor INs and A-MNs, animals exhibit exclusively reversallocomotion, with active body bending, robust rhythmicity, and velocity. Periodically, reversals were interrupted, when, with exaggerated head oscillation, the anterior and posterior body segments are pulled into opposing directions. (Parts 5, 6) Upon the co-ablation of premotor INs and D-MNs, animals exhibit kinker-like postures.

DOI: https://doi.org/10.7554/eLife.29915.006

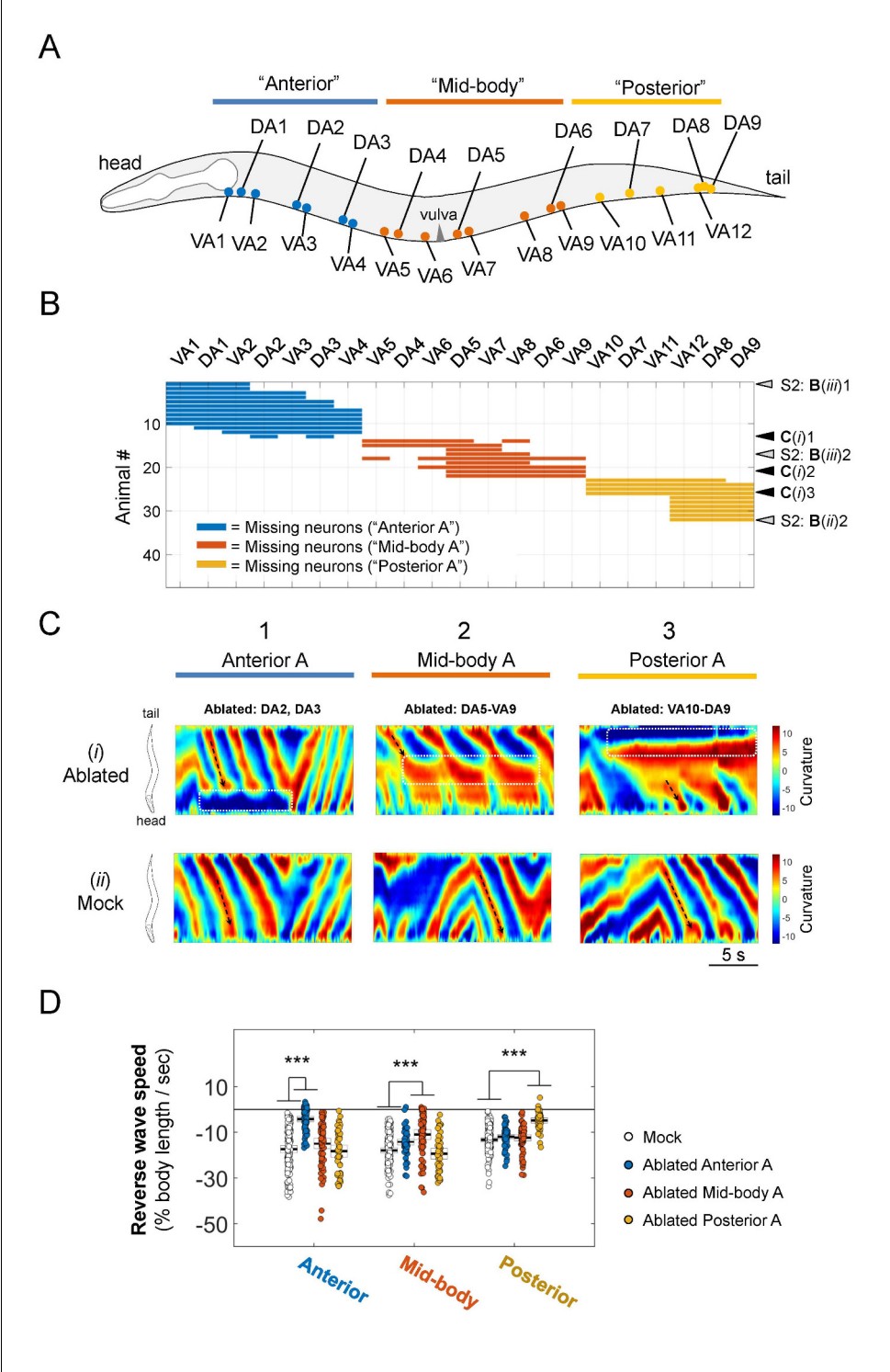

**Figure 3.** Sparse removal of A-MNs alters, does not abolish backward locomotion. (**A**) Schematic of the approximate locations of all A-MNs and regions of targeted ablation. An ablation is classified as 'Anterior', 'Mid-body', or 'Posterior' when at least one neuron from each region was ablated, and no neurons from other regions were ablated. (**B**) Missing A-MNs for each animal that was classified as Anterior (*n* = 13), Mid-body (*n* = 9), Posterior (*n* = 10), or Mock (*n* = 17) ablated. Black and gray arrowheads denote animals whose curvature kymograms are shown in (**C**) and ***Figure 3—figure supplement 1***, respectively. (**C**) Representative curvature kymograms for each ablation type (upper panels) and mock controls for three strains from which pooled ablation data were quantified (lower panels). (**D**) The rate of reversal bending wave propagation in the anterior, mid- and posterior body for each ablation class. Each dot represents one bout of reversal movement > 3 s. Black bars indicate the mean, and white boxes denote the 95%

*Figure 3 continued on next page*

*Figure 3 continued*
confidence interval of the mean. Ablation decreases bending frequency locally, but not in other body regions. \*\*\*p<0.001 by one-way ANOVA followed by Bonferroni post-hoc comparisons.
DOI: https://doi.org/10.7554/eLife.29915.007
The following figure supplement is available for figure 3:

**Figure supplement 1.** Information on all other partial A-MN-ablated animals.
DOI: https://doi.org/10.7554/eLife.29915.008

activities suffice for organized bending propagation, A-MNs exhibited phase relationships consistent with the temporal activation pattern of their muscle targets. The dorsal muscle-innervating DA7 exhibited anti-phasic activity change with that of ventral muscle-innervating VA10 and VA11 (*Figure 5B,C*), whereas the activity change of the more posterior muscle-innervating VA11 preceded that of VA10 (*Figure 5B,C*). Their oscillation frequency, positively correlated with the crawling speed, typically fell below 50 s (see *Figure 5B* for a sample trace and quantifications in later sections).

The difference between the frequency of A-MN-dependent rPSC bursts in dissected preparations, and A-MN calcium oscillation in immobilized versus crawling *C. elegans*, similar to what has been reported for the spinal cord preparations versus immobilized or mobile vertebrates (*Goulding, 2009*) may reflect the effect of motor and proprioceptive feedback.

## A-MN's oscillatory activity is potentiated by optogenetically activated premotor INs

A-MNs receive synaptic inputs from several premotor INs, most prominently from a pair of descending INs AVA. The AVA INs make both chemical and electrical synapses with all A-MNs (*White et al., 1986*). Optogenetic stimulation of AVA activates and maintains backward locomotion, and induces rPSCs in dissected preparations (*Gao et al., 2015*).

These results raise the possibility that premotor IN activation can promote the reversal motor state through potentiating A-MNs' oscillatory activitives. Indeed, AVA activation-induced rPSCs exhibited similar amplitude and frequency to A-MN-dependent endogenous rPSC bursts, albeit with less variability (*Figure 6A*). Importantly, upon the removal of A-MNs, AVA-evoked rPSC bursts were abolished (*Figure 6B*); this effect was not observed when either B-MNs, or their input premotor INs AVB were ablated (*Figure 6B–D*). Therefore, both evoked and intrinsic rPSC bursts in these preparations primarily originated from A-MNs, consistent with stimulated descending premotor IN inputs potentiating A-MN's intrinsic oscillation.

## Evoked and intrinsic A-MN oscillations require a P/Q/N-type VGCC

Because A-MN's oscillatory activity was robustly evoked by optogenetic stimulation of premotor INs, we used this preparation to screen for potential cation channels that underlie locomotor CPG's intrinsic membrane oscillation (*Harris-Warrick, 2002*). We examined three channels known to be expressed by MNs, the P/Q/N-type VGCC (*Mathews et al., 2003*; *Schafer and Kenyon, 1995*), the L-type VGCC (*Lee et al., 1997*), and the sodium leak channel (*Xie et al., 2013*).

rPSC bursts were readily evoked in mutants containing a partial loss-of-function (*lf*) allele for the pore-forming α-subunit of the L-VGCC CaV1α EGL-19 (*Figure 7—figure supplement 1*), as well as in mutants without the sodium leak channel's pore-forming NCA-1 and NCA-2 (*Gao et al., 2015*), and the auxiliary UNC-79 and

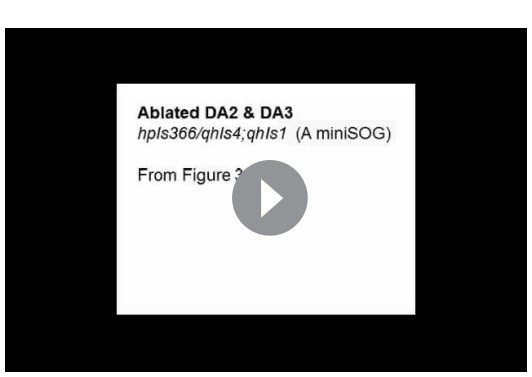

**Video 3.** Local ablation of some A-MNs does not prevent body bends in other segments. During reversals, localized ablations of a fraction of the A-MNs lead to defective local bends, but do not abolish bending in other segments. Example movies for the behavioral consequence of ablating anterior, mid-body, and posterior A-MNs are shown.
DOI: https://doi.org/10.7554/eLife.29915.009

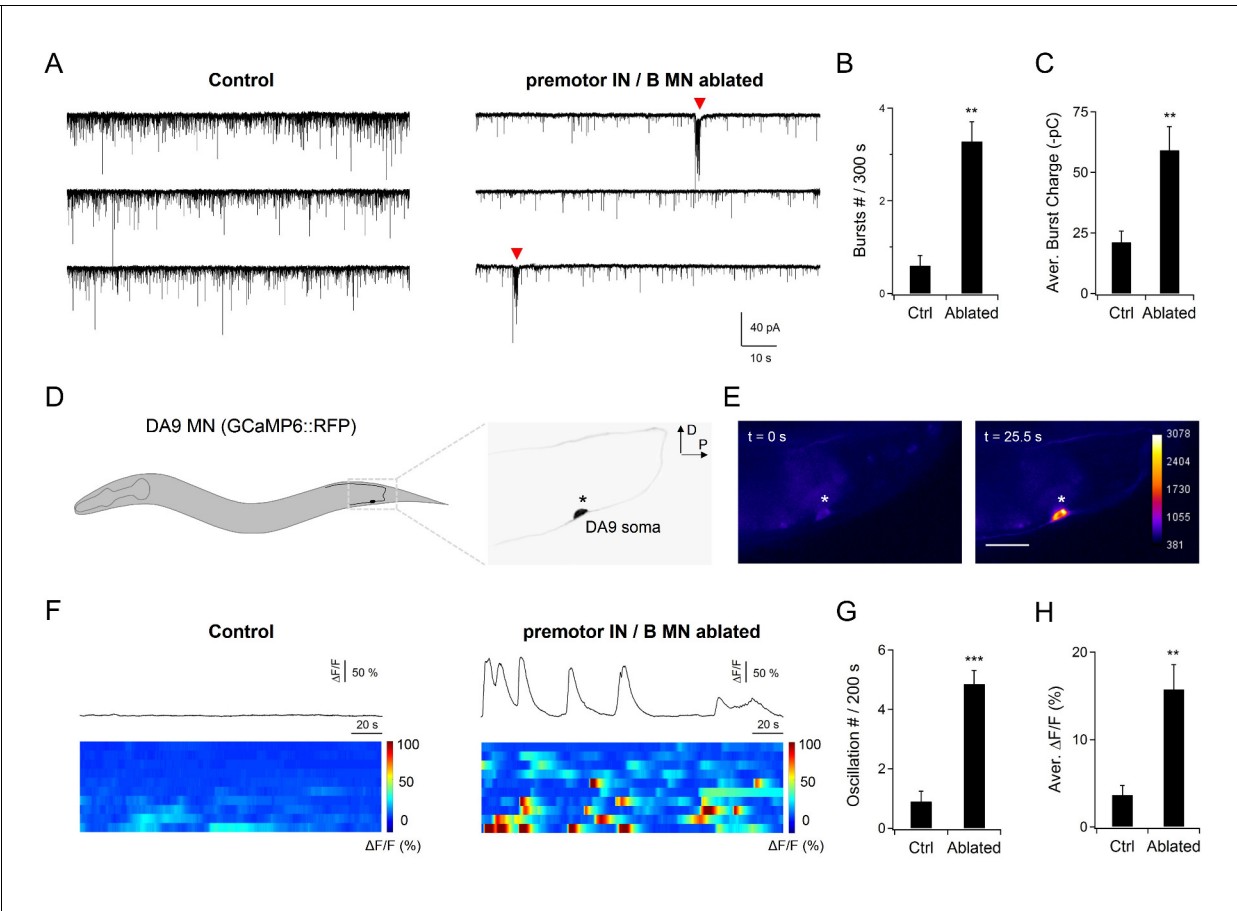

**Figure 4.** A-MNs exhibit rhythmic activities upon the ablation of premotor INs . (**A**) A representative post-synaptic PSC recording at the NMJ preparation of the same genotype, without (Control, left panel) or with (Ablated, right panel) LED illumination-induced ablation of premotor INs and B-MNs. Rhythmic PSC burst events (arrowheads) were consistently observed upon the removal of premotor INs and B-MNs. (**B, C**) Quantification of the rPSC burst frequency (**B**) and discharge (**C**), without (Ctrl) or with (Ablated) ablation. Both the rPSC bursts frequency and discharge are significantly increased in Ablated animals. $n$ = 10 animals each group. (**D**) Schematics of the morphology and trajectory of the DA9 MN soma and processes, visualized by an A-MN GCaMP6s::RFP $Ca^{2+}$ reporter. (**E**) Fluorescent signals during oscillatory $Ca^{2+}$ changes in the DA9 soma. (**F**) Examples of the DA9 soma $Ca^{2+}$ transient traces, and raster plots of recordings from individual animals of the same genotype, without (Control) and with (Ablated) the ablation of premotor INs and B-MNs. $n$ = 10 animals each group. (**G, H**) Quantification of the $Ca^{2+}$ oscillation frequency and mean total $Ca^{2+}$ activities, without (Ctrl) and with (Ablated) the ablation of premotor INs and B-MNs. Both the oscillation frequency and total activity of DA9 are significantly increased in ablated animals. **p<0.01, ***p<0.001 against the respective non-ablated Control group by the Mann-Whitney U test. Error bars, SEM.
DOI: https://doi.org/10.7554/eLife.29915.010

The following figure supplements are available for figure 4:

**Figure supplement 1.** Rhythmic postsynaptic action potential bursts were observed upon the co-ablation of premotor INs and B-MNs at the neuromuscular preparation.
DOI: https://doi.org/10.7554/eLife.29915.011

**Figure supplement 2.** Direct optogenetic stimulation of A-MNs leads to rhythmic PSCs.
DOI: https://doi.org/10.7554/eLife.29915.012

UNC-80 (*Figure 7—figure supplement 1*) subunits. By contrast, in a partial *lf* mutant for the α-subunit of the P/Q/N-VGCC CaV2α UNC-2, AVA stimulation failed to evoke rPSC bursts, despite increasing the mPSC frequency (*Figure 7—figure supplement 1*). We observed similar effects in mutants that lack auxiliary subunits of the P/Q/N-type VGCC, UNC-36 and CALF-1 (*Saheki and Bargmann, 2009*) (*Figure 7—figure supplement 1*). The specific loss of evoked rPSC bursts implicates a requirement of the P/Q/N-type VGCC for A-MN's intrinsic oscillatory activity. Indeed, endogenous rPSC bursts, which we observed upon the removal of all premotor INs and B-MNs in wildtype animals, were also diminished in *unc-2(lf)* mutants (*Figure 7A–C*).

## The P/Q/N-type VGCC underlies A-MN oscillation independently of synaptic transmission

UNC-2 is expressed exclusively in neurons. Like the vertebrate P/Q- and N-type VGCCs, it mediates presynaptic calcium influx, subsequently activating synaptic vesicle fusion and neurotransmitter release (*Mathews et al., 2003*). The loss of endogenous and evoked rPSC bursts in *unc-2*, *unc-36*, and *calf-1 lf* mutants implicate three possibilities: first, UNC-2-conducted high-voltage-activated calcium current is part of A-MN intrinsic membrane oscillation; second, A-MN' oscillation is dependent on synaptic transmission between premotor INs and A-MNs; third, in these mutants, A-MNs oscillate, but reduced synaptic transmission between premotor INs and A-MNs, and/or between A-MNs and muscles led to reduced rPSCs.

We can distinguish these possibilities by examining the calcium activity of A-MN somata. Specifically, we compared DA9's activity in intact but immobilized wildtype and *unc-2(lf)* animals, after co-ablating premotor INs and B-MNs. Devoid of premotor IN inputs, wildtype DA9 exhibited periodic calcium oscillations (*Figure 7D–F*, wild type). In *unc-2(lf)* mutants, both the amplitude and frequency of these oscillations were severely reduced (*Figure 7D–F*, *unc-2(e55; lf)*). Restoration of UNC-2 in A-MNs alone mutants was sufficient to restore oscillation of DA9's calcium signals in *unc-2(lf)* mutants (*Figure 7—figure supplement 2*). These results argue that independent of its role in exocytosis, UNC-2-mediated high-voltage-activated calcium current is an intrinsic component of A-MN membrane oscillation.

In this case, genetic impairment of exocytosis should not result in A-MN oscillatory defects. UNC-13 is a conserved and essential effector of presynaptic calcium influx to trigger exocytosis and neurotransmitter release (*Brose et al., 1995*; *Gao and Zhen, 2011*; *Richmond and Jorgensen, 1999*). Strikingly, in *unc-13(lf)* near null mutants, when premotor INs and B-MNs were ablated, DA9 exhibited periodic calcium oscillations as in wildtype animals (*Figure 7G–I*). This result not only confirms that UNC-2 has an exocytosis-independent function, but also reinforces the notion that A-MNs generate and sustain oscillatory activity in the absence of chemical synaptic input.

Lastly, if UNC-2 directly contributes to A-MN membrane oscillation, it should exhibit physical presence outside the presynaptic termini. We examined the subcellular localization of UNC-2 by inserting GFP at the endogenous *unc-2* locus (Materials and methods). Indeed, in addition to the presynaptic localization along the neuronal processes in both the central (*Figure 8E*, nerve ring) and peripheral (*Figure 8E*, VNC) nervous system (*Saheki and Bargmann, 2009*; *Xie et al., 2013*), punctate signals decorate the plasma membrane of neuron somata, including those of A-MNs (*Figure 8E*), consistent with UNC-2 conducting extracellular calcium influx in somata.

## Altering A-MN's oscillatory property leads to changes in the reversal velocity and propensity

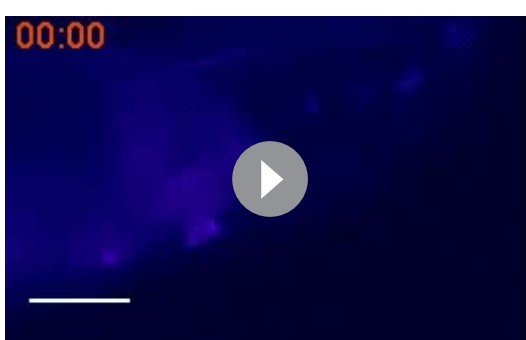

**Video 4.** The DA9 soma exhibits robust Ca²⁺ oscillation upon the co-ablation of all premotor INs and B-MNs. An example video of Ca²⁺ oscillation at the DA9 motor neuron in an animal without premotor INs and B-MNs.
DOI: https://doi.org/10.7554/eLife.29915.013

For channels that produce membrane oscillation, mutations that alter their kinetics should lead to corresponding changes in properties of oscillators and behaviors.

Consistent with this notion, *unc-2(lf)* mutants exhibited drastically reduced A-MN- dependent rPSC bursts (*Figure 7A–C*), and DA9's calcium oscillation was reduced in both amplitude and frequency (*Figure 7D–F*). We further isolated and examined the effect of *unc-2* gain-of-function (*gf*) mutations (Materials and methods) that reduce the channel inactivation kinetics, resulting in prolonged channel opening (Huang and Alkema; to be submitted; Alcaire and Zhen, unpublished). In contrast to the case of *unc-2(lf)* mutants, upon ablation of premotor INs and B-MNs, *unc-2(gf)* exhibited endogenous rPSC bursts with strikingly higher frequency and amplitude than wildtype animals (*Figure 7A–C*). DA9's calcium oscillation

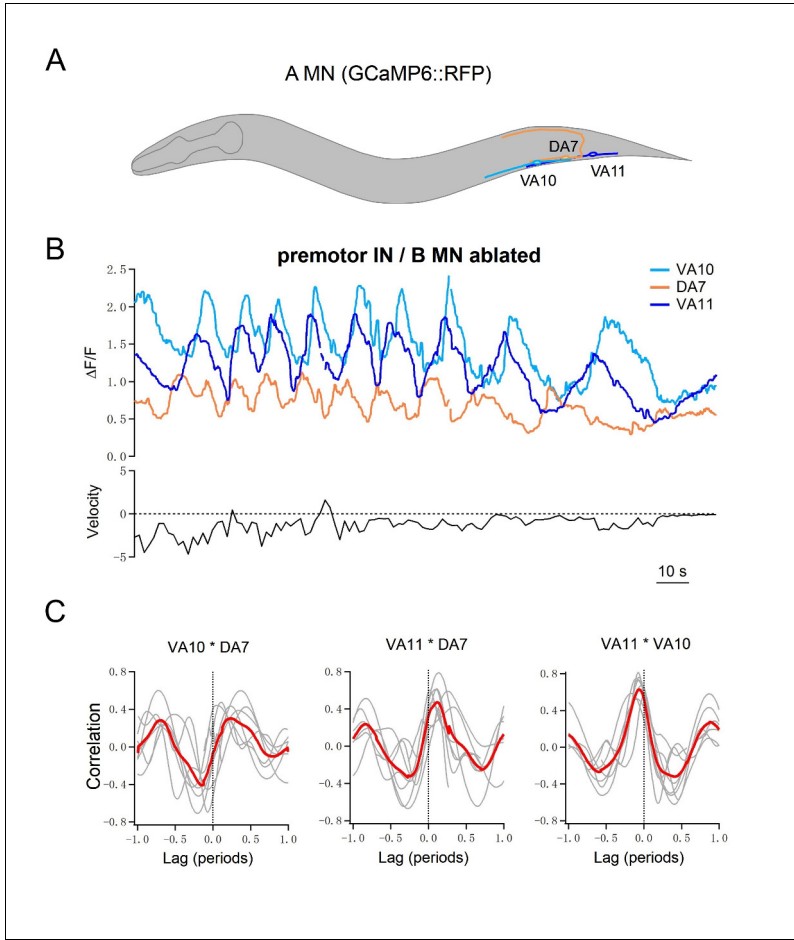

**Figure 5.** Multiple A-MNs are phase coupled to generate backward locomotion in the absence of premotor INs and B-MNs. (**A**) Schematics of the morphology and trajectory of the VA10, DA7, VA11 MN somata and processes. (**B**) *Upper panel:* example traces of calcium activities of three A-MNs, in animals where all premotor INs and B-MNs have been ablated. The VA10 and VA11 innervate adjacent ventral body wall muscles; the DA7 innervates dorsal muscles in opposition to those by the VA10 and VA11. Periodic Ca$^{2+}$ oscillations were observed in all neurons, represented by changes in the GCaMP6/RFP ratio (Y-axis) over time (X-axis). *Lower panel:* the animal's instantaneous velocity (Y-axis) simultaneously recorded during calcium imaging, was represented by the displacement of the VA11 soma. Values above and below 0 indicate forward (displacement toward the head) and backward (displacement toward the tail) locomotion, respectively. This animal exhibited continuous reversals. Note that the speed of Ca$^{2+}$ change during oscillation correlates with reversal velocity. (**C**) Phasic relationships among DA7, VA10, and VA11. DA7's activity change is anti-phasic to that of VA10 and V11, whereas the activity changes of VA10 and VA11 exhibit a small phase shift, with VA11 preceding VA10. The red line denotes the mean of all recordings. *n* = 7 animals.

DOI: https://doi.org/10.7554/eLife.29915.014

exhibited drastically increased frequency and amplitude in *unc-2(gf)* animals when compared to wild-type animals (*Figure 7D–F*). Moreover, in *unc-2(lf)* mutants, specific restoration of wildtype UNC-2 (WT) or Gain-of-function UNC-2(GF) in A-MNs was sufficient to rescue the frequency and amplitude of DA9's oscillation (*Figure 7—figure supplement 2*).

Altered A-MN oscillatory property corresponded with changes in reversal velocity and propensity. Upon the ablation of premotor INs and B-MNs, *unc-2(lf)* and *unc-2(gf)* mutants exhibited substantially reduced and increased reversal velocities, respectively, than ablated wildtype animals (*Figure 8A–C*). The decrease and increase of reversal velocity was accompanied by a decreased and increased propensity for backward locomotion, respectively (*Figure 8D*).

Therefore, not only UNC-2 conducts currents that support A-MN's membrane oscillation, modification of A-MN oscillatory property by either decreasing or increasing UNC-2's activity, is sufficient to alter the property of backward locomotion.

### Descending INs utilize a mixed synapse configuration to exert state-dependent inhibition and potentiation of A-MN oscillation to regulate the reversal motor state

When intrinsic A-MN activity drives backward locomotion, premotor IN inputs can control the reversal motor state by regulating their activities. The AVA descending premotor INs make both gap junctions and chemical synapses to all A-MNs (*Kawano et al., 2011*; *Liu et al., 2017*; *Starich et al., 2009*; *White et al., 1986*). Previous studies showed that AVA exert state-dependent and opposite effects on reversals: at rest, AVA-A gap junctions reduce spontaneous reversal propensity (*Kawano et al., 2011*), whereas optogenetic stimulation of AVA results in sustained backward locomotion (*Gao et al., 2015*; *Kato et al., 2015*). Therefore, AVA may control the reversal motor state through dual regulation - inhibition and potentiation - of A-MN oscillatory activity through the mixed gap junction and chemical synapse configuration.

To address whether at rest, AVA-A gap junctions dampen A-MN's intrinsic activity, we examined an *unc-7* innexin null mutant, in which gap junction coupling between AVA and A-MNs is disrupted (*Kawano et al., 2011*; *Liu et al., 2017*). DA9 exhibited low-calcium activity in both wildtype animals (*Figure 9D*) and the chemical synaptic transmission defective *unc-13(lf)* mutants (*Figure 7G*), in which AVA-A coupling remains intact. By contrast, DA9 exhibited robust calcium oscillation in *unc-7 (lf)* mutants, despite the presence of AVA (*Figure 9D*). When UNC-7 was restored specifically in AVA (Materials and methods), DA9's calcium oscillation was dampened in *unc-7(lf)* mutants (*Figure 9D–F*). When premotor INs and B-MNs were ablated, robust DA9 calcium oscillations were observed across the wildtype animals (*Figure 4F*), *unc-13(lf)* (*Figure 7G–I*), and *unc-7(lf)* mutants (*Figure 9—figure supplement 1A–C*), confirming that A-MN's intrinsic oscillatory property was unaltered by the *unc-7* mutation. Together, these results demonstrate that descending INs AVA inhibit A-MN's intrinsic oscillation via gap junction coupling.

Such an inhibition reduces propensity for the reversal motor state. As reported (*Kawano et al., 2011*; *Starich et al., 2009*), *unc-7* mutants display a *kinker* posture and drastically increased spontaneous reversals, in contrast to wildtype animal's strong bias for forward locomotion (*Figure 9G,H*; *Figure 9—figure supplement 1D*). When UNC-7 was restored specifically in the AVA premotor INs (Materials and methods), the *unc-7* mutant's spontaneous reversals were inhibited, accompanied by a restored bias for forward locomotion (*Figure 9G,H*). Their forward velocity was lower than that of wildtype animals (*Figure 9I*), which may reflect UNC-7's presence (*Altun et al., 2009*; *Starich et al., 2009*), hence potential requirement in other neurons to fully rescue the *unc-7* mutant's motor defects. As expected, after premotor INs and B-MNs ablation, the *unc-7* mutant exhibited continuous backward locomotion (*Figure 9—figure supplement 1D,F*).

Consistent with the notion that activated AVA potentiate A-MN oscillation, optogenetic activation of AVA evoked robust A-MN-dependent rPSC bursts (*Figure 6B–D*), and sustained reversals (*Gao et al., 2015*). AVA potentiate A-MNs primarily through chemical synapses, because AVA-evoked rPSC bursts exhibited normal frequency, with a modestly reduced total charge transfer in the *unc-7* mutant (*Figure 9A–C*). As expected, rPSCs were abolished in a chemical synaptic transmission defective mutant *unc-13* (*Figure 9A–C*). Thus, AVA's dual action – attenuation and potentiation – on the reversal motor state correlates with an inhibition and stimulation of A-MN's oscillatory activity, respectively.

In summary, we show that A-MNs exhibit intrinsic and oscillatory activity that is sufficient for backward locomotion. Positive and negative regulation of their oscillatory activity, through either manipulation of an endogenous oscillatory calcium current, or, by altering the synaptic inputs of descending premotor INs AVA, lead to changes in the propensity, velocity, and duration of the reversal motor state (*Figure 10*).

## Discussion

We show that multiple oscillators underlie *C. elegans* locomotion. For the motor circuit that promotes reversals, the excitatory A-MNs are both motor executors and rhythm generators. Through a

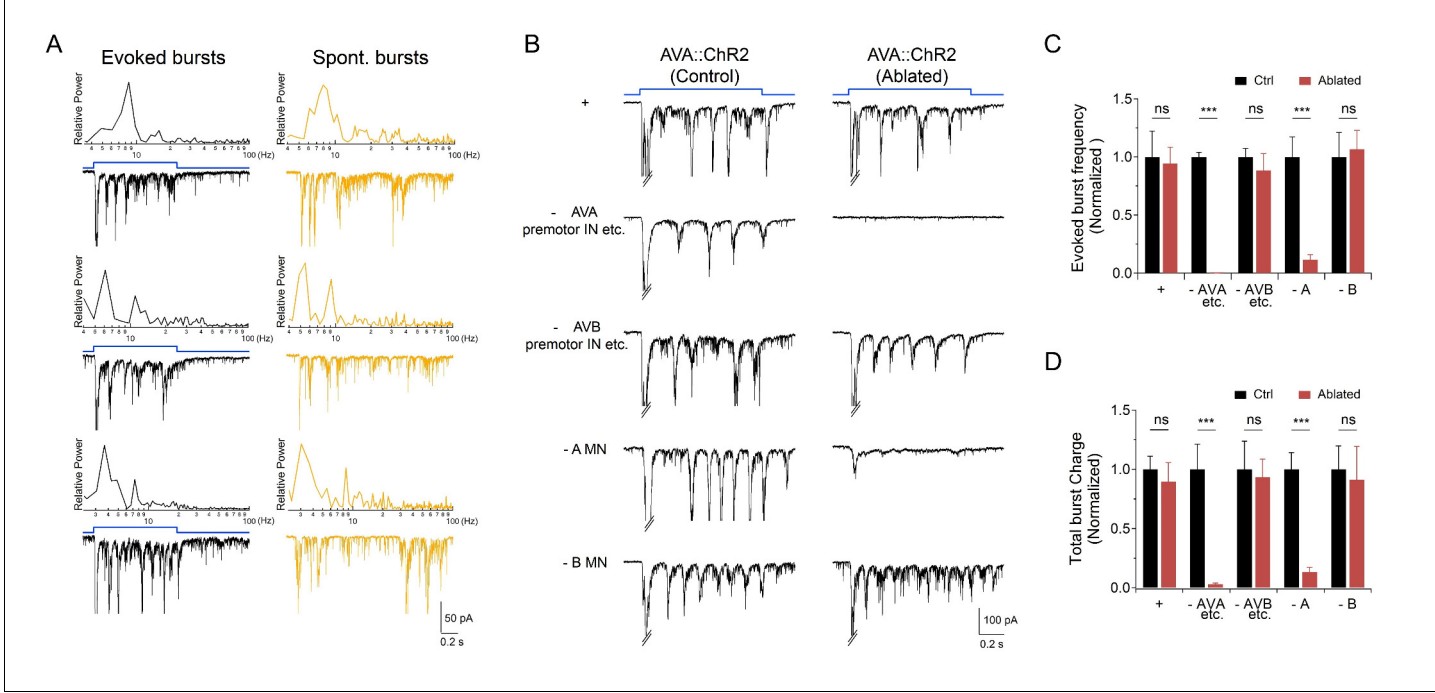

**Figure 6.** Activation of premotor INs AVA potentiates A-MN-dependent rPSC bursts. (**A**) The evoked and spontaneous rPSC bursts share frequency spectrum characteristics. *Black traces*: frequency spectrum analyses (upper panel) for three rPSC traces upon the optogenetic activation of AVA premotor INs (lower panel); *Yellow traces*: frequency spectrum analyses (upper panel) for three spontaneous rPSC bursts exhibited by animals upon the co-ablation of premotor INs and B-MNs (lower panel). (**B**) Representative traces of AVA-evoked rPSC bursts in respective genotypic backgrounds, both in the presence (Control) or absence (Ablated) of specific neuronal groups. +: *hpIs270* (AVA-specific ChR2 activation upon exposure to LED, in wildtype background); - AVA: *hpIs270; hpIs321* (upon expsure to LED, after a subset of premotor INs including AVA were ablated); - AVB: *hp270; hpIs331* (upon exposure to LED, after several INs including AVB were ablated); - A: *hpIs270; hpIs371* (upon exposure to LED, after A-MNs were ablated); - B: *hpIs270; hpIs604* (upon exposure to LED, after B-MNs were ablated). (**C**) Quantification of rPSC burst frequencies evoked by AVA activation in respective genetic backgrounds. (**D**) Quantification of total discharge of rPSC bursts evoked by AVA in respective genetic backgrounds. Both are diminished upon the ablation of AVA, but are not affected by ablation of the AVB premotor INs. They are both significantly decreased in A-, but not B-MN-ablated animals ($n \geq 5$ in each data set). ns, not significant ($p > 0.05$), ***$p < 0.001$ against the respective non-ablated Control group by the students' $t$ - test. Error bars, SEM.

DOI: https://doi.org/10.7554/eLife.29915.015

mixed synapse configuration, the descending AVA premotor INs exert state-dependent inhibition and activation of their intrinsic activity to regulate the reversal motor state. We discuss that results from this and previous studies reconcile previously conceived differences on the electrophysiological properties and anatomic organization between *C. elegans* and other locomotor networks. We further discuss that these results exemplify circuit compression: the *C. elegans* motor circuit operates with fundamental similarities to large locomotor networks, but with fewer neurons and circuit layers.

## Separate oscillators of different properties drive forward and backward locomotion

*C. elegans* locomotion involves the orchestration of at least three independent rhythm generators that underlie the head and body movements. Forward locomotion consists of head oscillations that pull the body forward and body oscillations that are driven by B-MNs. Backward locomotion consists of body oscillations that are driven by multiple A-MNs, and may be antagonized by the head oscillation.

Separating the forward- and reversal-driving oscillators at the motor neuron level offers a different strategy for directional movements. In the lamprey and *Drosophila* larvae, the same motor neuron group underlies body movements; a directional change is achieved through reversing the direction of propagation.

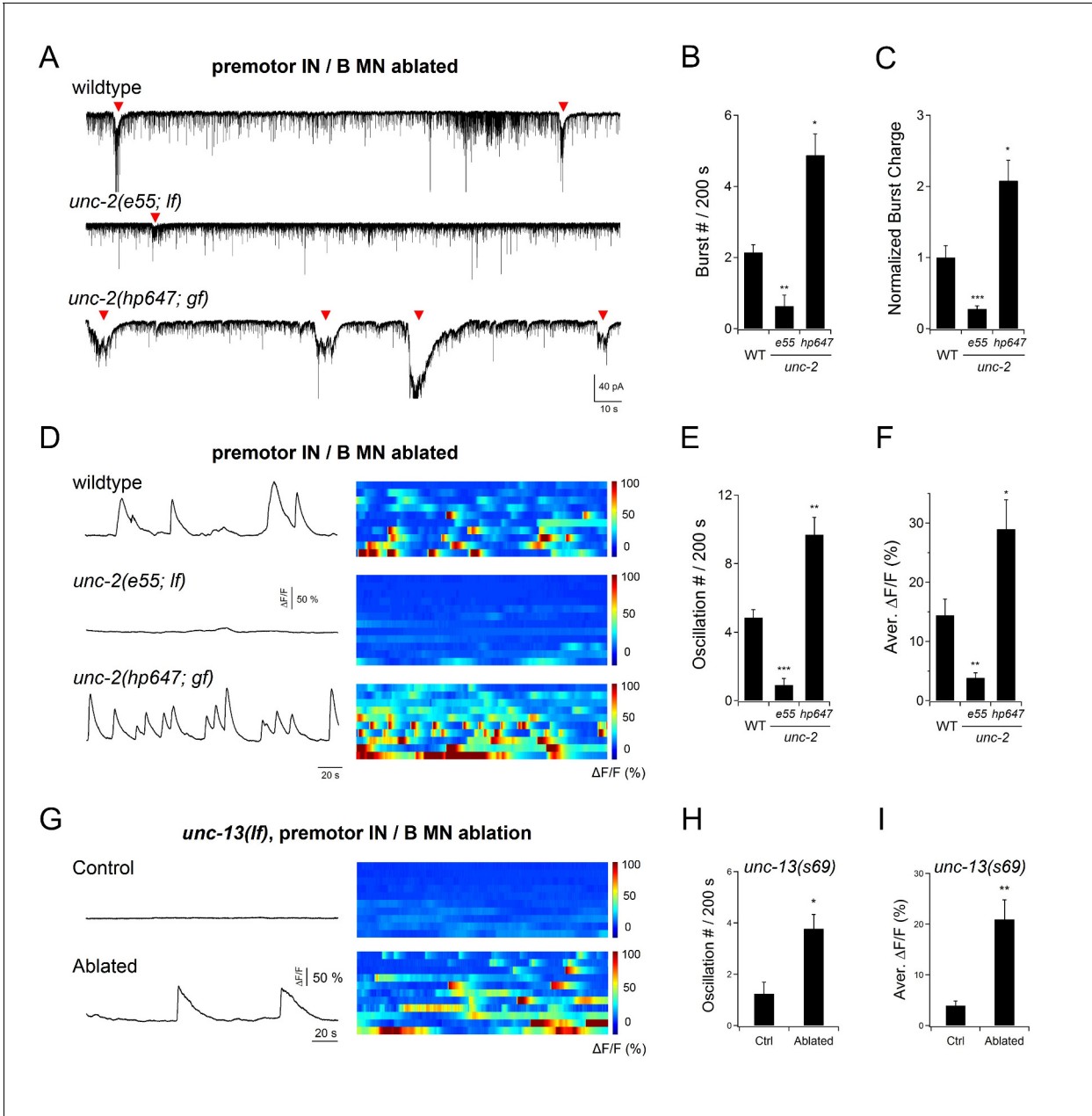

**Figure 7.** An endogenous UNC-2 channel activity regulates A-MN's rhythmic activity. (A) Representative PSC recordings of the NMJ preparations of respective genotypes, after the co-ablation of premotor INs and B-MNs. The amplitude and frequency of periodic rPSC bursts (arrowheads) were reduced in *unc-2(e55; lf)*, and increased in *unc-2(hp647; gf)* mutant animals. (B) Quantification of the rPSC burst frequency in respective genotypes. (C) Quantification of the total discharge of rPSC bursts in respective genotypes. Both were reduced in *unc-2(e55; lf)*, and increased in *unc-2(hp647; gf)* mutants ($n \geq 7$ in each dataset). (D) Example traces of the DA9 soma $Ca^{2+}$ recording (left panels), and raster plots of all $Ca^{2+}$ recordings (right panels) in wildtype animals ($n = 10$), *unc-2(e55; lf)* ($n = 12$), and *unc-2(hp647; gf)* ($n = 10$) mutants upon the co-ablation of premotor INs and B-MNs. (E) Quantification of DA9's $Ca^{2+}$ oscillation frequency in respective genotypes upon the co-ablation of premotor INs and B-MNs ablation. (F) Quantification of the overall DA9 $Ca^{2+}$ activity in respective genotypes upon premotor INs and B-MNs. When compared to wildtype animals, both the frequency and activity of $Ca^{2+}$ oscillation are significantly reduced in *unc-2(e55; lf)* and increased in *unc-2(hp647; gf)* mutant animals. (G) Example DA9 soma $Ca^{2+}$ traces (left panels) and raster plots of all $Ca^{2+}$ traces (right panels) in *unc-13(lf)* mutants, without (Control, $n = 10$) and with (Ablated) the co-ablation of premotor INs and B-MNs (Ablated, $n = 11$). (H, I) Quantification of the $Ca^{2+}$ oscillation frequency (H) and overall activities (I) in *unc-13* mutants. *$p<0.05$, **$p<0.01$, ***$p<0.001$ against the respective non-ablated Control group by the Mann-Whitney U test.
DOI: https://doi.org/10.7554/eLife.29915.016

The following figure supplements are available for figure 7:

**Figure supplement 1.** The P/Q/N-type VGCC UNC-2 is required for evoked rPSC bursts.

*Figure 7 continued on next page*

*Figure 7 continued*

DOI: https://doi.org/10.7554/eLife.29915.017

**Figure supplement 2.** Cell-autonomous UNC-2 conductance underlies DA9 calcium oscillation.

DOI: https://doi.org/10.7554/eLife.29915.018

In the absence of premotor INs, *C. elegans* generates deeper body bends during backward than forward locomotion. This suggests that reversal-driving A-MNs have intrinsically higher activity than the forward-promoting B-MNs. This predicts potential differences in their relationships with the respective regulatory premotor INs. Indeed, establishing the forward movement as a preferred motor state requires the descending premotor INs of the reversal-promoting motor circuit AVA to attenuate A-MN's activity through electrical coupling (*Kawano et al., 2011*; this study). We speculate that motor neurons with high intrinsic activities are advantageous for reversal, a motor state that is frequently incorporated in adverse stimuli-evoked behaviors such as escape and avoidance.

## A-MNs are rhythm generators

The following evidence supports that motor neurons that execute backward locomotion are themselves the oscillators: they exhibit intrinsic and oscillatory activity; their intrinsic activity is sufficient to drive backward locomotion; their oscillatory frequency is positively correlated with the reversal velocity; premotor IN-mediated attenuation and potentiation of their activities determine the propensity and maintenance of the reversal motor state.

In other locomotor circuits, MNs are thought to provide feedbacks to the rhythm generating premotor INs (*Heitler, 1978*; *Song et al., 2016*; *Szczupak, 2014*). This study describes the first example of MNs performing both motor and rhythm-generating roles in a locomotor circuit. These findings bear resemblance to the lobster pyloric circuit, where two pyloric MNs, both capable of intrinsic membrane oscillation and bursting, are integral components of an oscillatory network that underlies constitutive food filtering (*Marder and Bucher, 2001*; *Selverston and Moulins, 1985*).

## High-voltage-activated calcium currents are a conserved component of oscillation

In addition to a role in exocytosis, the P/Q/N-type calcium conductance produces A-MN membrane oscillation. In *unc-2(lf)*, but not *unc-13(lf)* mutants in which synaptic transmission was abolished (*Richmond and Jorgensen, 1999*), A-MN calcium oscillation was compromised. Endogenously tagged UNC-2 resides at both the presynaptic termini and soma of MNs. Restoring UNC-2 in A-MNs in *unc-2(lf)* – simultaneously rescuing their oscillation and NMJ activities – restores backward locomotion in the absence of premotor INs.

An exocytosis-independent role of high-voltage-activated calcium channels in membrane oscillation has been noted in vertebrates. In isolated lamprey spinal neuron somata, the N-type calcium currents prominently potentiate action potential bursts, and are coupled with calcium-activated potassium currents that terminate the bursts (*el Manira et al., 1994*; *Wikström and El Manira, 1998*). The intrinsic, high-frequency gamma band oscillation of the rat pedunculopontine nucleus (PPN) requires high-threshold N- and/or P/Q-type calcium currents, a finding that is consistent with dendritic and somatic localization of VGCC channels in cultured PPN neurons (*Hyde et al., 2013*; *Kezunovic et al., 2011*; *Luster et al., 2015*). Voltage-activated calcium conductance may be a shared property of neurons with oscillatory activity.

## Calcium-driven oscillators and bursters underlie *C. elegans* motor rhythm

*C. elegans* is superficially at odds with several fundamental features of CPG-driven networks: its genome does not encode voltage-activated sodium channels that typically drive action potential bursts, and all examined neurons to date, MNs and premotor INs included, are non-spiking. Instead, MNs and premotor INs exhibit plateau potentials upon stimulation (*Kato et al., 2015*; *Liu et al., 2014*).

Results from this and previous studies reveal a simplified, but fundamentally conserved cellular and molecular underpinning of rhythmicity in *C. elegans* locomotion. B- and A-MNs exhibit calcium

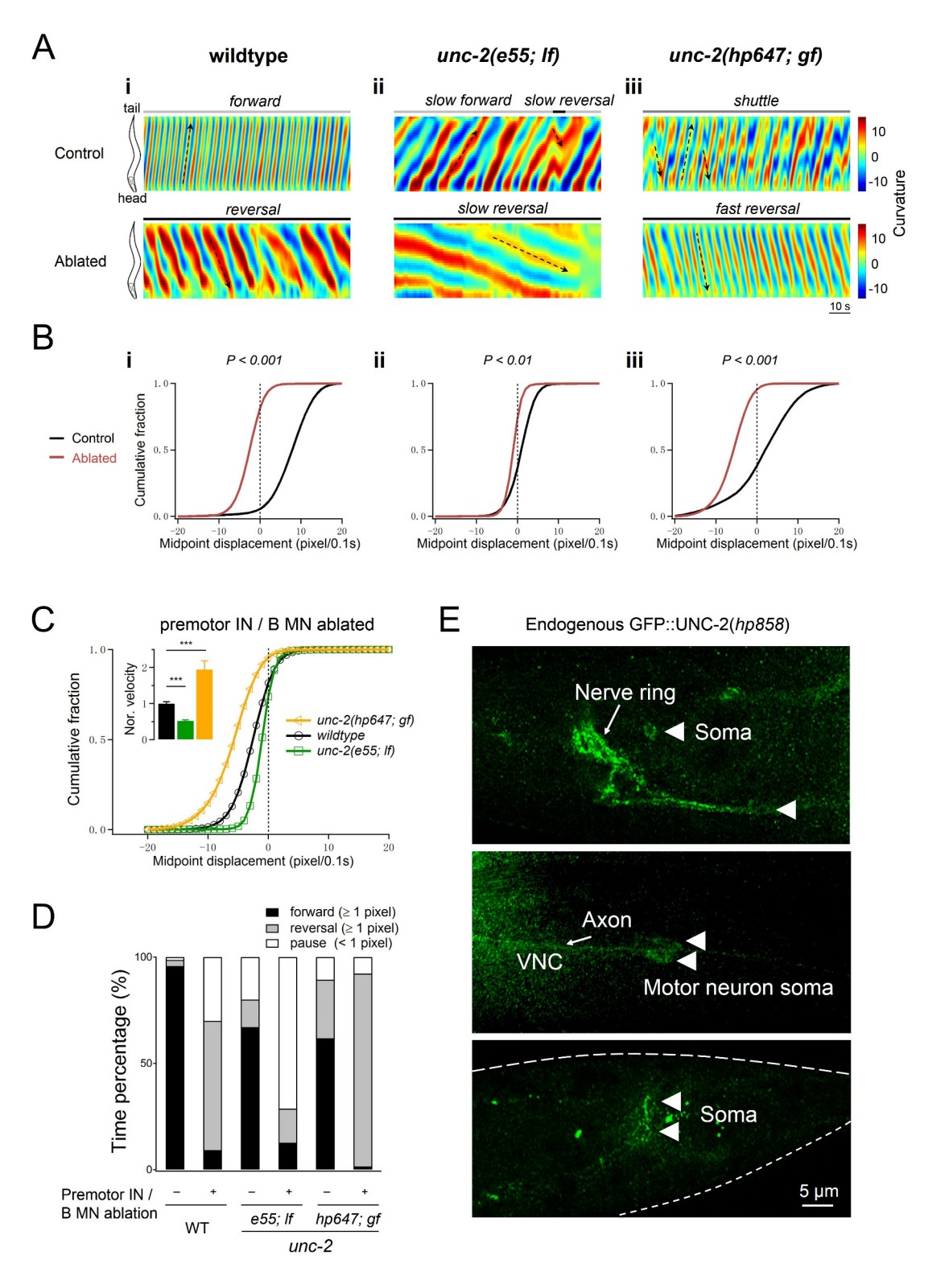

**Figure 8.** Increased UNC-2 activity leads to increased reversal velocity and duration. (**A**) Representative curvature kymograms of wildtype and *unc-2* mutant animals, without (Control) and with (Ablated) the co-ablation of premotor INs and B-MNs. Black arrows on kymograms denote directions of bending propagation. (**B**) Distribution of instantaneous velocity exhibited by animals of respective genotypes, presented by the animal's mid-point displacement where the positive and negative values represent the forward and backward locomotion, respectively. While all animals exhibit backward

*Figure 8 continued on next page*

*Figure 8 continued*

locomotion upon the co-ablation of premotor INs and B-MNs, the speed of bending wave propagation, representing the reversal velocity, is significantly reduced and increased in *unc-2(lf)* and *unc-2(gf)* mutants, respectively. *n* = 10 animals per group. **p<0.01, ***p<0.001 against the non-ablated animals of the same genotype by the Kolmogorov-Smirnov test. (C) Distribution of the instantaneous velocity of wildtype, *unc-2(lf)*, and *unc-2 (gf)* mutant animals, upon the removal of premotor INs and B-MNs. Decreased UNC-2 activity leads to drastic reduction of velocity, whilst increased UNC-2 activity leads to increased velocity. ***p<0.001 against premotor IN and B-MN-ablated wildtype animals by the Kolmogorov-Smirnov test. Error bars, SEM. (D) Propensity of directional movements in animals of respective genotypes, quantified by the animal's mid-point displacement. Upon the removal of premotor INs and B-MNs, all animals shift to a bias for backward locomotion; Note that *unc-2(lf)* mutants also exhibit a significant increase of pauses, whereas *unc-2(gf)* mutants eliminate forward locomotion. (E) The expression pattern of endogenous UNC-2, determined by a GFP::UNC-2 (*hp858*) knock-in allele, stained with antibodies against GFP. Dense, punctate signals decorate the nerve processes of the central and peripheral nervous systems, as well as somata in central nervous system (top panel) and ventral cord motor neurons (middle panel), including the DA8 and DA9 soma (bottom panel). VNC, ventral nerve cord. Scale bar: 5 μm.

DOI: https://doi.org/10.7554/eLife.29915.019

oscillations (*Kawano et al., 2011*; *Wen et al., 2012*; this study). Muscles alone fire calcium-driven action potentials (*Gao and Zhen, 2011*; *Jospin et al., 2002*; *Liu et al., 2011*, *2009*; *Raizen and Avery, 1994*). Activation of premotor INs or MNs triggers rhythmic bursting in body wall muscles, a pattern of physical relevance (*Gao et al., 2015*). Further, not only are A-MNs required for muscle bursting; altering A-MN oscillation changes the frequency and duration of bursting (this study).

We propose that the *C. elegans* locomotor network utilizes a combinatory oscillatory and bursting property of MNs and muscles to generate motor rhythm. In the absence of voltage-activated sodium channels, high-voltage-activated calcium channels, specifically, the UNC-2 P/Q/N- and L-type VGCCs that drive MN oscillation and muscle spiking, respectively, take deterministic roles in the rhythmic output.

## Functional and anatomic compression at the *C. elegans* motor circuit

In the spinal cords, distinct pools of spinal premotor INs and MNs play dedicated roles in rhythm generation, pattern coordination, proprioceptive and recurrent feedback, and execution of different motor patterns (*Grillner, 2006*; *Kiehn, 2016*). *C. elegans* operates with a remarkably small number of neurons. Despite the numeric simplicity, *C. elegans* exhibits remarkable adaptability and repertoire of motor behaviors. Its motor infrastructure has to compress multiple functions into a smaller number of cells and fewer layers of neurons.

Indeed, body wall muscles are the only bursting cells. Excitatory MNs absorb the role of rhythm generators. Previous (*Wen et al., 2012*) and this study suggest that *C. elegans* MNs are likely proprioceptive. The anatomy, where only the MN somata reside at the ventral nerve cord, supports the notion that *C. elegans* MNs and muscles perform all essential functions of the large locomotor CPG networks including the spinal cords.

Anatomical constraints may have necessitated functional compression. When a nervous system consists of a small number of neurons, instead of being simple, it is more compact. The numeric complexity, reflected by both increased neuronal subtypes, numbers and layers in large circuits, is compensated by a cellular complexity that endows individual neuron or neuronal class multi-functionality in a small nervous system.

## A mixed synapse configuration regulates dynamics of rhythm-generating circuits

Mixed chemical and electrical synaptic connectivity between INs and MNs may be a universal feature of rhythm-generating circuits. Not only do locomotor circuits in the crayfish, leech, *C. elegans*, Drosophila (*Matsunaga et al., 2017*), and zebrafish exhibit this conserved configuration, a similar organization has been found in other motor systems, including the lobster cardiac and stomatogastric ganglia (*Hartline, 1979*; *Marder, 1984*), and the snail feeding system (*Staras et al., 1998*). Prevalent gap junctions have also been reported at the mature rodent spinal cords (*Kiehn and Tresch, 2002*).

In several preparations, gap junctions allow MNs to retrogradely regulate the activity of premotor INs (*Heitler, 1978*; *Liu et al., 2017*; *Song et al., 2016*; *Staras et al., 1998*). This, however, may be an overly simplified interpretation on their physiological functions. When arranged in combination

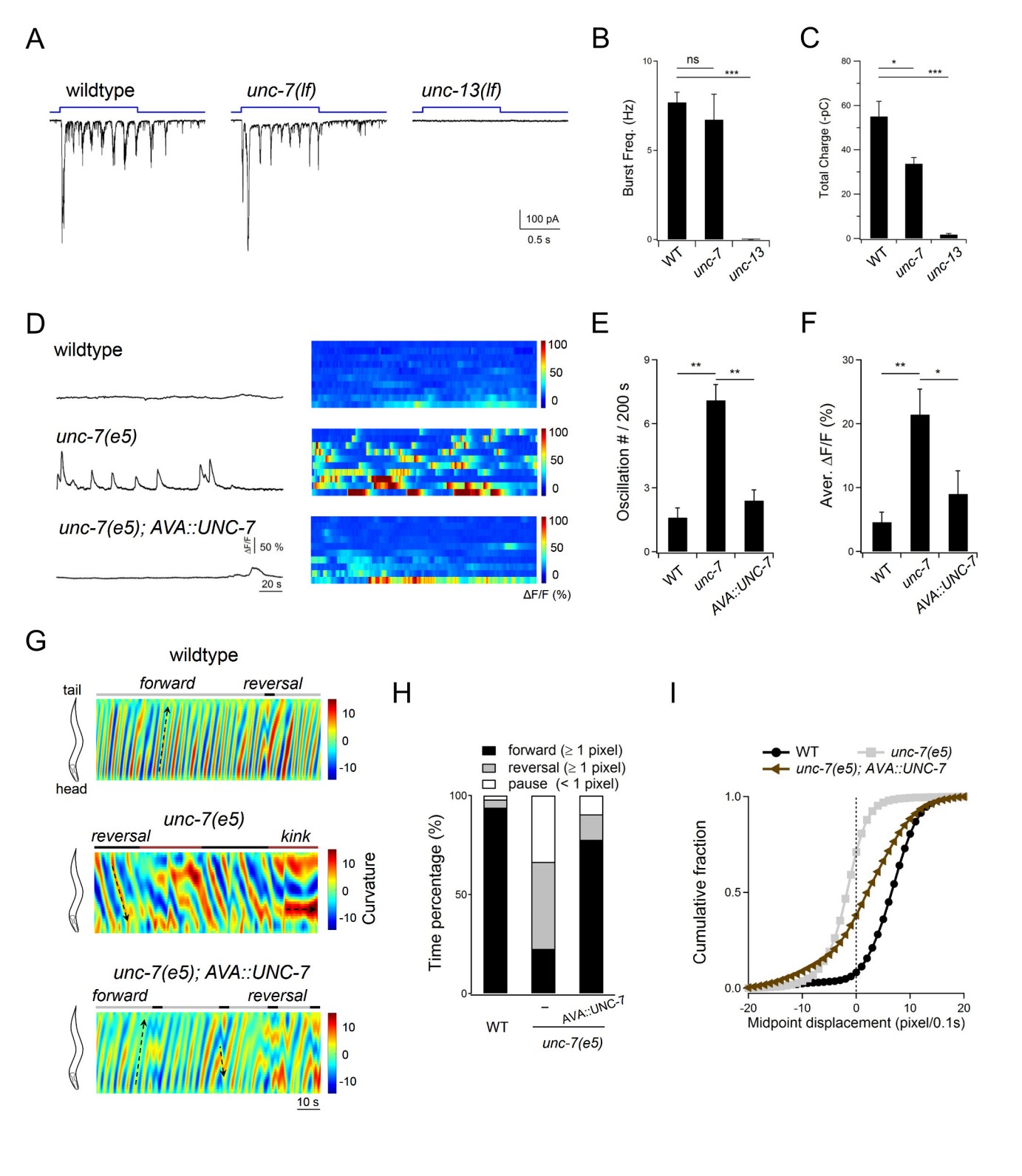

**Figure 9.** Descending premotor INs AVA exert dual modulation - inhibition and potentiation - of A-MN's oscillatory activity to control the reversal motor state. (A) Representative rPSC recordings in wildtype, *unc-7(lf)* and *unc-13(lf)* animals upon the optogenetic stimulation of premotor INs AVA. (B, C) Quantification of the frequency (B) and total discharge (C) of rPSC bursts in respective genotypes. *n* = 16, 7 and 3 animals for wildtype, *unc-7* and *unc-13*, respectively. (D) Representative DA9 soma Ca²⁺ traces and raster plots of all Ca²⁺ traces in wildtype, *unc-7(lf)* and AVA-specific UNC-7-rescued

*Figure 9 continued on next page*

*Figure 9 continued*

animals, all with the presence of premotor INs and B-MNs. *n* = 10 animals each group. (**E, F**) Quantification of the $Ca^{2+}$ oscillation frequency (**E**) and mean overall activities (**F**) in respective genotypes. DA9's activity exhibits significant increase in *unc-7(lf)* mutants in the presence of premotor INs and B-MNs. *P<0.05,**p<0.01, ***p<0.001 against the rspective wildtype Control group by the Mann-Whitney U test. Error bars, SEM. (**G**) Representative curvature kymograms of wildtype, *unc-7(lf)* and AVA-specific UNC-7-rescued animals in the presence of premotor INs and B-MNs. Upward and downward pointing black arrows on kymograms denote posterior- and anterior-propagating body bends; the horizontal black arrow denotes the absence of propagation. (**H**) Propensity of directional movements in animals of respective genotypes, quantified by the animal's mid-point displacement. A specific restoration of UNC-7 in AVA significantly rescued *unc-7* mutant animal's bias for backward movement and pause. (**I**) Distribution of instantaneous velocity of respective genotypes, presented by the animal's mid-point displacement. AVA-specific UNC-7 expression also partially rescued the mobility of *unc-7(lf)* mutant. *n* = 10 animals per group.
DOI: https://doi.org/10.7554/eLife.29915.020

The following figure supplement is available for figure 9:

**Figure supplement 1.** DA9 oscillation is not changed in *unc-7* mutant animals after the removal of premotor INs and B-MNs.
DOI: https://doi.org/10.7554/eLife.29915.021

with chemical synapses, gap junctions can exert diverse effects on circuit dynamics and flexibility (*Marder et al., 2017*; *Rela and Szczupak, 2004*). In the crustacean's pyloric network, the Anterior Burster (AB) IN, Pylorid Dilator (PD) MN, and Ventricular Dilator (VD) MN are electrically coupled. Mixed electrical coupling and inhibitory chemical synapses between AB and VD allows VD and PD MNs to fire out-of-phase despite their electrical coupling (*Marder, 1984*). In the gastric network, a descending interneuron MCN1 tonically activates the LG gastric mill neurons and DG dorsal gastric neurons; a mixed electrical coupling and reciprocal chemical synapse wiring between LG and MCN1 contribute to the alternate firing of LG and DG (*Coleman et al., 1995*). In the leech swimming circuit, mixed electrical coupling and inhibitory chemical synapses between premotor INs and MNs facilitate recurrent inhibition on MNs (*Szczupak, 2014*). A full understanding of their physiological functions requires examination of their impact on behaviors.

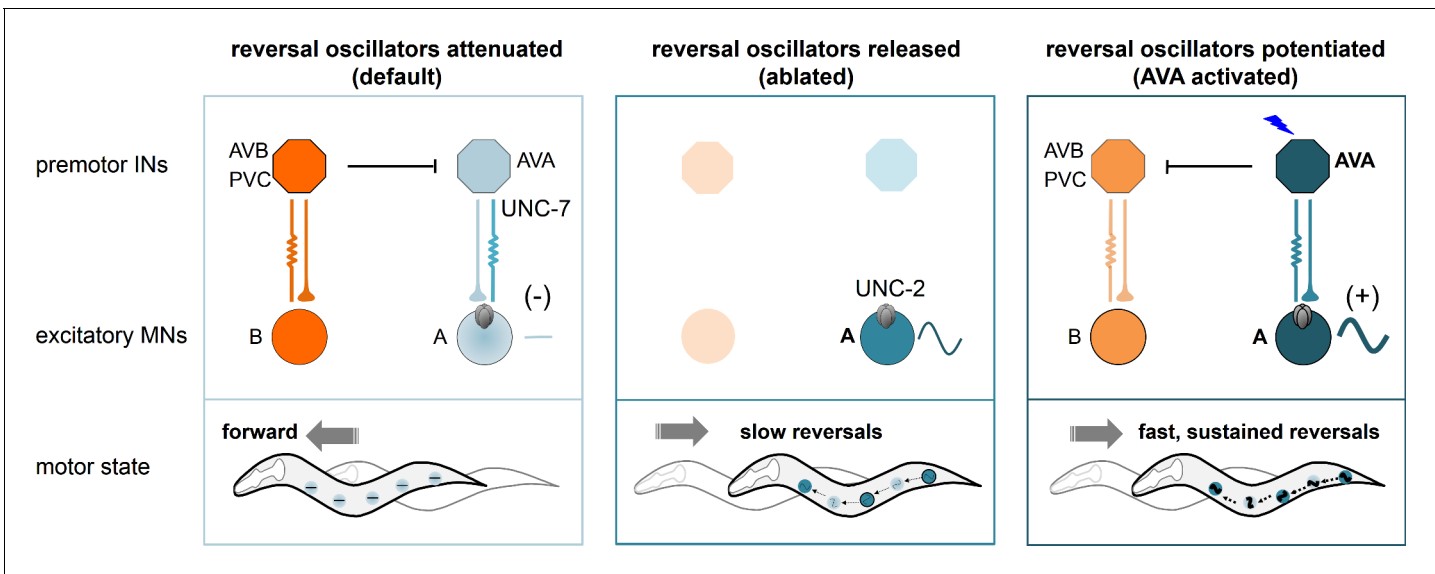

**Figure 10.** A model for a distributed reversal oscillator-driven motor circuit, dually regulated by the descending premotor INs to modulate the reversal motor state. Multiple A-MNs represent distributed and phase-coordinated intrinsic oscillators to drive backward locomotion. The descending premotor INs AVA exert state-dependent dual regulation - inhibition and activation – on A-MN's oscillatory activity to determine the initiation and substation of the reversal motor state through a mixed electrical and chemical synapse configuration. Left panel: at rest, A-MN intrinsic activity is inhibited by AVA through UNC-7 innexin-dependent coupling. Center panel: the ablation of premotor INs removes AVA-A coupling, releasing UNC-2 VGCC-dependent A-MN calcium oscillation, and permitting slow backward locomotion. Right panel: upon stimulation, AVA potentiate A-MN's oscillatory activity, primarily through chemical synapses, with a minor contribution from electrical synapses, permitting sustained reversals.
DOI: https://doi.org/10.7554/eLife.29915.022

*C. elegans* innexin mutants have permitted direct behavioral assessment of the role of gap junctions. Combined with electrophysiological and optogenetic examination, innexin mutants that selectively disrupt premotor IN and MN gap junctions (*Kawano et al., 2011*; *Liu et al., 2017*; this study) reveal sophisticated roles for a mixed, heterotypic and rectifying gap junction and excitatory chemical synapse configuration in locomotion.

In the reversal motor circuit, this configuration allows the AVA premotor INs to exert state-dependent inhibition and activation on its oscillators. At rest, AVA-A gap junctions dampen the excitability of premotor INs (*Kawano et al., 2011*), and the oscillatory activity of A-MNs (this study) to reduce the propensity for spontaneous reversals. Upon activation, AVA potentiate A-MNs predominantly through chemical synapses, with a minor contribution from gap junctions (this study). The weakly rectifying gap junctions (*Liu et al., 2017*; *Starich et al., 2009*) may allow activated A-MNs to antidromically amplify the excitatory chemical synaptic inputs from AVA, prolonging evoked reversals (*Liu et al., 2017*). Because genetic studies for gap junctions are lacking in most experimental systems, we may continue to be surprised by the sophistication and diversity of such a configuration.

## Remaining questions

Our analyses indicate that A-MNs function as a chain of phase-coupled local oscillators to organize and execute backward locomotion. The following questions should be addressed to verify and refine this model.

First, how A-MNs are phase-coupled to execute backward locomotion in the absence of premotor interneurons. Forward-promoting B-MNs are strongly activated by proprioception (*Wen et al., 2012*). In the reversal-promting circuit, proprioception may, as in other motor circuits, serve as feedbacks to regulate A-MN oscillation and to organize their phasic relationship. Such a model proposes that A-MNs integrate the role of not only rhythm generating, but also proprioceptive INs in other locomotor circuits.

Second, which molecular mechanisms underlie the multiple identities of excitatory MNs. Comparing the intrinsic difference between properties of A- and B-MNs, and potentially among individual members of each MN class, provides a starting point to evaluate this hypothesis.

Third, the molecular identities of other channels, in addition to UNC-2, that endow A-MN oscillatory membrane property. Previous studies characterized membrane properties of lamprey locomotor oscillators pharmacologically: the depolarization initiated by sodium and calcium conductance, potentiated and maintained by voltage- or glutamate-activated calcium conductance, and terminated by calcium- and voltage-activated potassium currents (*Grillner et al., 2001*). *C. elegans* genetics should allow us to reveal the precise molecular identity and channel composition. This pursuit may further delineate mechanisms that underlie circuit compression.

Fourth, in the lamprey spinal cord, the fastest oscillator entrains other oscillators and leads propagation (*Grillner, 2006*). DA9, the most posterior unit that exhibits the highest activity, poises to be a leading oscillator of the *C. elegans* reversal motor circuit. A-MNs serve as a genetic model to address circuit and molecular mechanisms that endow the property of the leading CPG, and underlie the entrainment and coupling of trailing oscillators.

Lastly, as in all other motor systems, A-MN oscillators must coordinate with those that drive other the motor states. Circuit and molecular mechanisms that underlie their coordination must be addressed in this system.

## Closing remarks

Our studies contribute to a growing body of literature that small animals can solve similar challenges in organizing locomotor behaviors faced by larger animals, with a conserved molecular repertoire, and far fewer neurons. They serve as compact models to dissect the organizational logics of neural circuits, where all essential functions are instantiated, but compressed into fewer layers and cells.

# Materials and methods

## Constructs, transgenic arrays and strains

The complete lists of constructs, transgenic lines and strains generated or acquired for this study are provided in *Table 1* and *Table 2*. All *C. elegans* strains were cultured on the standard Nematode Growth Medium (NGM) plates seeded with OP50, and maintained at 15°C or 22°C. Unless stated otherwise, the wildtype animal refers to the Bristol N2 strain. Strains that contain optogenetic transgenes (*hpIs166, hpIs270, hpIs569* and *hpIs578*) were cultured in darkness on NGM plates supplemented with or without ATR (*Liewald et al., 2008*). Strains that contain miniSOG transgenes (*hpIs321, hpIs331, hpIs366, hpIs371, hpIs376, juIs440, hpIs583, hpIs590, hpIs481* and *hpIs603*) were maintained in darkness on NGM plates.

## Genetic mutants

*unc-2(hp647; gf)* was identified in a suppressor screen for the 'fainter' motor defects of *unc-80 (e1069)* mutants. Both *e1069; hp647* and *hp647* animals exhibit hyperactive locomotion with high movement velocity and frequent alternation between forward and backward locomotion (shuttling) (Alcaire and Zhen, unpublished results). *hp647* was mapped between *egl-17* and *lon-2* on the X chromsome. Co-injected fosmids WRM0628cH07 and WRM0616dD06 rescued the shuttling phenotype exhibited by *hp647* and reverted *unc-80(e1069); hp647* to *unc-80(e1069)*-like fainter phenotypes. Subsequent exon and genomic DNA sequencing revealed a causative, gf L653F mutation in *unc-2(hp647)*. *unc-2(hp858)* is an insertion allele where GFP was fused immediately in front of the ATG start codon of the *unc-2* locus by CRISPR. *hp858* animals exhibit wildtype motor behaviors. Other genetic mutants used for constructing transgenic lines and compound mutants were obtained from the *Caenorhabditis Genetics Center* (CGC); all were backcrossed at least four times against N2 prior to analyses.

## Constructs and molecular biology

All promoters used in this study were generated by PCR against a mixed-stage N2 *C. elegans* genomic DNA. Promoters include the 5.1 kb *Pnmr-1*, 4.8 kb *Prig-3*, 2.8 kb *Psra-11*, 1.8 kb *Pacr-2s*, 2.5 kb *Punc-4*, 4.2 kb *Pacr-5*, 0.9 kb *Pttr-39*, 2.8 kb *Pceh-12*, 2.7 kb *Punc-129(DB)*, and 0.86 kb *Plgc-55B* genomic sequence upstream of the respective ATG start codon. The *Pnmr-1* used in this study excluded a 2 kb internal fragment that encodes *cex-1*, which interferes with reporter expression (*Kawano et al., 2011*).

For calcium imaging constructs, the genetic calcium sensor GCaMP3 and GCaMP6s were used for muscle and neuronal calcium imaging, respectively. The GCaMP6s sequence (*Chen et al., 2013*) was codon-optimized for expression in *C. elegans*. The synthesized gene also contains three *C. elegans* introns and contain restriction enzyme sites to facilitate subsequent cloning. In all constructs, GCaMP6 was fused in frame with a *C. elegans* codon-optimized mCherry at the C-terminus to allow ratio metric measurement via simultaneous imaging of GFP and RFP.

For neuronal ablation constructs, MiniSOG (*Shu et al., 2011*) fused with an outer mitochondrial membrane tag TOMM20 (tomm20-miniSOG or mito-miniSOG) (*Qi et al., 2012*) was used. An intercistronic sequence consisting of a U-rich element and Splice Leader sequence (UrSL) was inserted between the coding sequence of tomm20-miniSOG and Cherry or BFP to visualize neurons that express miniSOG and the efficacy of ablation. Inter-cistronic region consisting of a U-rich element and Splice Leader sequence (UrSL) between *gpd-2* and *gpd-3* was PCR amplified with OZM2301 (AAGCTAGCGAATTCGCTGTCTCATCCTACT TTCACC) and OZM2302 (AAGGTACCGATGCG TTGAAGCAGTTTC CC) using pBALU14 as the template. Two sets of bicistronic expression reporters used in this study, codon-optimized mCherry and EBFP, were gifts of Desai (University of California, San Diego) and Calarco (University of Toronto), respectively. They were used for behavioral analyses and to be combined with calcium imaging analyses, respectively.

The FRT-FLP system was used to generate two constructs that are co-injected to achieve AVA-specific UNC-7 expression. *Prig-3* was used to drive the expression of a UNC-7::GFP cassette, and *Pnmr-1* was used to drive FLP expression.

**Table 1.** Constructs and transgenic arrays generated or acquired for this study.

| Purpose | Plasmid | Description | Host strain/ Injection marker | Transgene | Strain |
|---|---|---|---|---|---|
| Neuron Ablation | pJH2843 | *Punc-4*::tomm20::miniSOG-SL2::RFP (A-MNs/others)* | *lin-15/Lin-15* | *hpIs366** | ZM7690* |
| | | | | *hpIs371** | ZM7691* |
| | pJH2842 | *Pacr-5*::tomm20-miniSOG-SL2::RFP (B-MN/others) | *lin-15/Lin-15* | *hpIs372* | ZM7798 |
| | pJH2844 | *Punc-25*::tomm20-miniSOG-SL2::RFP (D-MN/others) | *lin-15/Lin-15* | *hpIs376* | ZM7696 |
| | pJH2827 | *Pnmr-1*::tomm20-miniSOG-SL2::RFP (AVA/E/D/others) | *lin-15/Lin-15* | *hpIs321* | ZM7054 |
| | pJH2890 | *Plgc55B*::tomm20-miniSOG-SL2::RFP (AVB/others) | *lin-15/Lin-15* | *hpIs331* | ZM7297 |
| | † | *Psra-11*::tomm20-miniSOG/ *Psra-11*::RFP (AVB/others) | † | *juIs440*† | CZ19093† |
| | pJH3626 | *Pacr-2(s)*::tomm20-miniSOG-SL2::RFP (A-/B-MN-specific) | *lin-15/Lin-15* | *hpIs583* | ZM9062 |
| | pJH3458 | *Pttr-39*::tomm20-miniSOG-SL2::BFP (D-MN-specific) | *lin-15/Lin-15* | *hpIs590* | ZM9123 |
| | pJH3409/pJH3410 | *Pceh-12*::tomm20-miniSOG-SL2::BFP/ *Punc-129(DB)*::tomm20-miniSOG-SL2::BFP (B-MN-specific) | *lin-15/Lin-15* | *hpIs481* | ZM8607 |
| | pJH3449/pJH3453/ pJH3456 | P*lgc-55B*::tomm20-miniSOG-SL2::BFP/ P*nmr-1*::tomm20-miniSOG-SL2::BFP/ P*acr-5*::tomm20-miniSOG-SL2::BFP (all premotor IN/B MN/others) | *lin-15/Lin-15* | *hpIs603* | ZM9176 |
| Neuron ID | pJH1841 | *Pacr-2*::RFP | *dpy-20/Dpy-20* | *qhIs4* | YX146 |
| Optogenetic Stimulation of Neurons | pJH1697 | *Pglr-1*::ChR2(H134R)::YFP (AVA/AVE/AVD/others) | *lin-15/Lin-15* | *hpIs166* | ZM4624 |
| | pJH2690/pJH2673 | *Prig-3*::FRT ::ChR2(H134R)::RFP/ *Pnmr-1*::FLP (AVA-specific) | *lin-15/Lin-15* | *hpIs270* | ZM6804 |
| | pJH3558 | *Punc-4*::Chrimson::Cherry (A-MNs/others)* | *lin-15/Lin-15* | *hpIs569** | ZM8958* |
| | pJH3577 | *Pceh-12*::Chrimson::Cherry (VB-MNs) | *lin-15/Lin-15* | *hpIs578* | ZM8983 |
| UNC-2 Localization | pJH3931/pJH3932 | sgRNA and repair template for GFP insertion at the *unc-2* locus | N2 | *hp858* | ZM9583 |
| UNC-2 Rescue | pJH3916 | *Punc-4*::GFP::UNC-2 (WT) | *unc-2(e55); hpIs603; hpIs459/ Pmyo-2*::RFP | *hpEx3856* | ZM9495 |
| | pJH3917 | *Punc-4*::GFP::UNC-2 (*L653F; gf*) | *unc-2(e55); hpIs603; hpIs459/ Pmyo-2*::RFP | *hpEx3859* | ZM9498 |
| AVA-specific UNC-7 Rescue | pJH4119/pJH2673 | *Prig-3*::FRT::UNC-7::GFP/ *Pnmr-1*::FLP (AVA) | *unc-7(e5)/ Pmyo-3*::RFP | *hpEx3969* | ZM9816 |
| Calcium Imaging | pJH3137 | *Punc-4*::GCaMP6::RFP* (A-MNs/others) | *lin-15/Lin-15* | *hpIs459*[a] | ZM8428* |
| | | | | *hpIs460** | ZM8429* |
| | ‡ | *Pmyo-3*::GCaMP3::RFP (body wall muscles) | ‡ | *ljIs131*‡ | ZM7982‡ |

*In adults, *Punc-4* expression is more prominent in VC-MNs than A-MNs. We did not observe rhythmic calcium signals in VCs as in A-MNs (Gao and Zhen, unpublished). Mini-SOG ablation was performed in late L2 larvae; optogenetic stimulation in young L4 larvae, to ensure A-MN-specific ablation and activation.

†Acquired. Information about the construct, transgene and strain was described in Qi, Y. B. *et al.* Photo-inducible cell ablation in *Caenorhabditis elegans* using the genetically encoded singlet oxygen generating protein miniSOG. *Proc Natl Acad Sci U S A* 109, 7499–504 (2012).

‡Acquired. Information about the construct, transgene and strain was described in Butler, V. J. *et al*. A consistent Muscle activation strategy underlies crawling and swimming in *Caenorhabditis elegans*. *J R Soc Interface* 12, 20140963 (2015).
DOI: https://doi.org/10.7554/eLife.29915.023

## Transgenetic arrays and strains

Transgenic animals that carry non-integrated, extra-chromosomal arrays (*hpEx*) were generated by co-injecting an injection marker with one to multiple DNA construct at 5–30 ng/µl. Animals that carry integrated transgenic arrays (*hpIs*) were generated from the *hpEx* animals by UV irradiation, followed by outcrossing against N2 at least four times. L4-stage or young adults (24 hr post L4) hermaphrodites were used in all experiments.

## On plate whole population neuron ablation

To distinguish the role of different classes of neurons in locomotion modulation, we expressed mito-miniSOG into the A-, B-, D-MNs, premotor INs (AVA, AVE, PVC, AVD, AVB) and a few other unidentified neurons, respectively (*Table 1* and *Table 2*). On plate ablation of all members of MNs and premotor INs was performed using a homemade LED box, where the standard NGM culture plates, without lid, were exposed under 470 nm blue LED light box for 30–45 min. To monitor the specificity and efficacy of cell ablation, cytoplasmic RFP was co-expressed with miniSOG (tomm-20-miniSOG-SL2-RFP) in targeted neurons by the same promoter. Ablation was performed when most animals were in the L2 stage. Late L4 stage animals were recorded for behavioral analyses. Afterwards, they were mounted individually on agar pads to be examined for RFP signals; only recordings from animals where RFP signals were absent were analyzed.

## On plate locomotion analyses

A single 12–18 hr post-L4 stage adult hermaphrodite, maintained on standardculture conditions, was transferred to a 100 mm imaging plate seeded with a thin layer of OP50. One minute after the transfer, a 2-minute video of the crawling animal was recorded on a modified Axioskop 2 (Zeiss, Germany) equipped with an automated tracking stage MS-2000 (Applied Scientific Instruments, Eugene, OR) and a digital camera (Hamamatsu, Japan).

Imaging plates were prepared as follows: a standard NGM plate was seeded with a thin layer of OP50 12–14 hr before the experiment. Immediately before the transfer of worms, the OP50 lawn was spread evenly across the plate with a sterile bent glass rod. Movements exhibited by *C. elegans* were recorded using an in-house developed automated tracking program. All images were captured with a 10X objective at 10 frames per second. Data recorded on the same plate and on the same day were pooled, quantified and compared.

Post-imaging analyses utilized an in-house developed ImageJ Plugin (*Kawano et al., 2011*) and in-house written MATLAB scripts. Images from each animal were divided into 33 body segments. The mid-point was used to track and calculate the velocity and direction of movements between each frame. The percentage of total frames exhibiting pausing, backward or forward locomotion was defined by the mid-point displacement: between –1 pixel/second (- movements toward the tail) and + 1 pixel/second (+ movements toward the head) was defined as pause, less than –1 pixel/second backward, and more than + 1 pixel/s forward movement. The angle between three joint points was used to calculate the curvature of the middle point loci. The curvature value for each body segment was plotted, from the head to tail over time, and converted as a color map of movements.

## Calcium imaging of immobilized or crawling animals

DA9 recording from immobilized intact animals (all calcium imaging figures except *Figure 5*) was carried out as follows: animals were glued as described for electrophysiological recording (*Gao et al., 2015*), and imaged with a 60X water objective (Nikon, Japan) and sCMOS digital camera (Hamamatsu ORCA-Flash 4.0V2, Japan) at 100 ms per frame. Data were collected by MicroManager and analyzed by ImageJ.

Simultaneous imaging of multiple A-MNs (VA10, VA11, and DA7) in moving animals (*Figure 5*) was performed as described in a previous study (*Kawano et al., 2011*). Animals were placed on a 2% agarose pad on a slide, suspended in the M9 buffer, covered by a coverslip, and imaged with a

**Table 2.** Additional strains that were generated using the transgenic arrays in *Table 1*.

| Strain | Genotype | Purpose/Notes | Figure/Video |
|---|---|---|---|
| **For behavior analyses upon sparse A-MN ablation** | | | |
| YX167 | hpIs366/qhIs4; qhIs1* | * A transgene irrelevant to this study | *Figure 3*; *Video 3* |
| YX148 | qhIs4; qhIs1* | * A transgene irrelevant to this study | *Figure 3*; *Video 3* |
| **For behavior and electrophysiology analyses upon neuron ablation** | | | |
| ZM7971 | hpIs321; hpIs331 | All premotor IN ablation | *Figure 1*; *Video 1* |
| ZM9133 | hpIs583 hpIs589 | A-/B-/D-MN ablation | *Figure 1* |
| ZM8415 | hpIs321; juIs440 | All premotor IN ablation | *Figure 1—figure supplement 1* |
| ZM7862 | hpIs321; hpIs331; hpIs372 | All premotor IN/B-MN ablation | *Figure 2*; *Video 2*; *Figure 4* |
| ZM7870 | hpIs321; hpIs331; hpIs371 | All premotor IN/A-MN ablation | *Figure 2*; *Video 2* |
| ZM7921 | hpIs321; hpIs331; hpIs376 | All premotor IN/D-MN ablation | *Figure 2*; *Video 2* |
| ZM8668 | unc-2(e55; lf); hpIs321; hpIs331; hpIs372 | All premotor IN/B-MN ablation | *Figure 8* |
| ZM9022 | unc-2(hp647; gf); hpIs321; hpIs331; hpIs372 | All premotor IN/B-MN ablation | *Figure 8* |
| ZM9999 | unc-7(e5); hpIs603 | All premotor IN/B-MN ablation | *Figure 9—figure supplement 1* |
| **For Ca2²⁺imaging and/or behavior analyses** | | | |
| ZM7465 | hpIs321; hpIs331; ljIs131 | Muscle Ca²⁺ imaging, upon all premotor IN ablation | *Video 1* |
| ZM9228 | hpIs603; hpIs459 | A-MN Ca²⁺ imaging, upon all premotor IN/B-MN ablation | *Figures 4* and *5*; *Video 4* |
| ZM9229 | unc-2(e55; lf); hpIs603; hpIs459 | A-MN Ca²⁺ imaging and behavior, upon all premotor IN/B-MN ablation | *Figure 7* |
| ZM9495 | unc-2(e55; lf); hpIs603; hpIs459; hpEx3856 | A-MN Ca²⁺ imaging, UNC-2 (WT) rescue in A-MN | *Figure 7—figure supplement 2* |
| ZM9498 | unc-2(e55; lf); hpIs603; hpIs459; hpEx3859 | A-MN Ca²⁺ imaging, UNC-2 (GF) rescue in A-MN | *Figure 7—figure supplement 2* |
| ZM9231 | unc-2(hp647; gf); hpIs603; hpIs459 | Same as above | *Figure 7* |
| ZM9557 | unc-13(s69; lf); hpIs603; hpIs459 | Same as above | *Figure 7* |
| ZM8654 | unc-7(e5; lf); hpIs460 | Same as above | *Figure 9* |
| ZM9885 | unc-7(e5; lf); hpIs460; hpEx3966 | Same as above | *Figure 9* |
| ZM9657 | unc-7(e5; lf); hpIs603; hpIs460 | Same as above | *Figure 9—figure supplement 1* |
| **For optogenetic stimulation and electrophysiology analyses** | | | |
| ZM7920 | hpIs321; hpIs270 | AVA ChR2 stimulation, upon AVA/AVE/AVD/others ablation | *Figure 6* |
| ZM7861 | hpIs331; hpIs270 | AVA ChR2 stimulation, upon AVB/others ablation | *Figure 6* |
| ZM8408 | hpIs371; hpIs270 | AVA ChR2 stimulation, upon A-MN/others ablation | *Figure 6* |
| ZM9093 | hpIs481; hpIs166 | AVA/others ChR2 stimulation, upon B-MN/others ablation | *Figure 6* |
| ZM7615 | unc-2(e55; lf); hpIs166 | AVA/others ChR2 stimulation | *Figure 7—figure supplement 1* |
| ZM6646 | unc-2(ra612; lf); hpIs166 | AVA/others ChR2 stimulation | *Figure 7—figure supplement 1* |
| ZM7608 | unc-36(e251; lf); hpIs166 | AVA/others ChR2 stimulation | *Figure 7—figure supplement 1* |
| ZM7780 | calf-1(ky867; lf); hpIs166 | AVA/others ChR2 stimulation | *Figure 7—figure supplement 1* |
| ZM6646 | unc-79(e1068; lf); hpIs166 | AVA/others ChR2 stimulation | *Figure 7—figure supplement 1* |
| ZM7533 | unc-80(e1069; lf); hpIs166 | AVA/others ChR2 stimulation | *Figure 7—figure supplement 1* |
| ZM7509 | elg-19(n582; lf); hpIs166 | AVA/others ChR2 stimulation | *Figure 7—figure supplement 1* |
| ZM6639 | unc-13(e1091; lf); hpIs166 | AVA/others ChR2 stimulation | *Figure 9* |
| ZM4728 | unc-7(e5; lf); hpIs166 | AVA/others ChR2 stimulation | *Figure 9* |

DOI: https://doi.org/10.7554/eLife.29915.024

63X objective. Neurons were identified by their stereotypic anatomical organization. Multiple Regions Of Interest (ROI) containing the MN soma of interest were defined using an in-house developed MATLAB script. Videos were recorded with a CCD camera (Hamamatsu C2400, Japan) at 100 ms per frame. Simultaneous velocity recording at each time point was measured using an Image J plug-in developed in-house (*Gao et al., 2015*; *Kawano et al., 2011*).

In both preparations, GCaMP and RFP signals were simultaneously acquired using the Dual-View system (Photometrics, Tucson, AZ), and the GCaMP/RFP ratios were calculated to control for motion artifacts and fluorescence bleaching during recording. Late L4 stage animals were used for all calcium imaging experiments, when the *Punc-4* promoter exhibits strongest expression in A-MNs.

## Region-specific light-ablation of MNs and behavioral analyses

Data in *Figure 3* and *Figure 3—figure supplement 1* were collected from animals where A-type MNs were ablated using three strains, where A- or A/B-MNs were labeled by RFP. For miniSOG-based ablation, ZM9062 *hpIs583* (A- and B-MNs miniSOG) or YX167 *hpIs366/qhIs4; qhIs1* (A-MNs miniSOG) L2 larva were immobilized on a 4% agar pad with 5 mM tetramisole. Region-specific illumination was performed by targeting a 473 nm laser at an arbitrary portion of the animal using a digital micromirror device (DMD) through a 20X objective (*Leifer et al., 2011*). The final irradiance at the stage was approximately 16 mW/mm$^2$. The DMD was set to pulse the laser with a duty cycle of 1 s on, 0.8 s off, for a total of 300 of total ON time. Each animal was immobilized for a maximum of 30 min.

Most posterior MNs (VA12-DA9) ablation was also performed in YX148 *qhIs4; qhIs1* (A/B RFP) by a pulsed infrared laser illumination system (*Churgin et al., 2013*) modified with increased output power. L2 animals were immobilized in the same manner. A single 2 ms pulse was applied to each neuron through a 60X objective visualized by RFP. This procedure never affected VA11, the nearest non-targeted neuron. Following ablation, each animal was transferred to an OP50-seeded NGM plate and allowed to grow to the day 1 adult stage. Controls were animals of the same genotype treated identically except without blue or infrared laser illumination.

For behavior recordings, each animal was transferred to an unseeded NGM plate, and on plate crawling was recorded for at least 5 min under bright field illumination. If the animal became sluggish or idle, the plate was agitated using the vibration motor from a cell phone. After recordings, each animal was imaged at 40X for RFP pattern for the entire body and we manually assigned present and missing neurons based on their relative positions and commissural orientation (*White et al., 1976*; *White et al., 1986*). For YX167, where A- and B-MNs were labeled, two researchers independently analyzed the same image and discussed, and agreed on the identification. We only included data from animals where we were confident of the identity of neurons.

Analyses of locomotion of ablated and control animals were carried out using WormLab (MBF Bioscience, Williston, VT) and in-house Matlab codes. Data from all three ablation methods were pooled to generate the summary statistics. Bouts of reversals that lasted at least 3 s were analyzed for the speed of wave propagation. Curvature segmentations from the behavioral recordings were constructed using WormLab (MBF Bioscience, Williston, VT). Wave speed was measured as a function of body coordinate and time, by taking the derivative of each curvature map with respect to time (dκ/dt), and to body coordinate (dκ/dC). Wave speed was defined as the ratio between these gradients (body coordinate/s). Wave speed was averaged over the length of each bout, and binned for the anterior (5–25% of body length from the head), mid-body (40–60%), and posterior (75–95%) region in each bout.

## Electrophysiology and optogenetic stimulation

Dissection and recording were carried out using protocols and solutions described in (*Gao and Zhen, 2011*), which was modified from (*Mellem et al., 2008*; *Richmond and Jorgensen, 1999*). Briefly, 1- or 2-day-old hermaphrodite adults were glued (Histoacryl Blue, Braun, Germany) to a sylgard-coated cover glass covered with bath solution (Sylgard 184, Dowcorning, Auburn, MI). After clearing the viscera by suction through a glass pipette, the cuticle flap was turned and gently glued down using WORMGLU (GluStitch Inc., Canada) to expose the neuromuscular system. The integrity of the anterior ventral body muscle and the ventral nerve cord were visually examined via DIC microscopy (Eclipse FN1, Nikon), and muscle cells were patched using 4–6 MΩ-resistant borosilicate

pipettes (1B100F-4, World Precision Instruments, Sarasota, FL). Pipettes were pulled by micropipette puller P-1000 (Sutter, Novato, CA), and fire-polished by microforge MF-830 (Narishige, Japan). Membrane currents and action potentials were recorded in the whole-cell configuration by a Digidata 1440A and a MultiClamp 700A amplifier, using the Clampex 10 and processed with Clampfit 10 software (Axon Instruments, Molecular Devices, Sunnyvale, CA). Currents were recorded at holding potential of –60 mV, while action potentials were recorded at 0 pA. Data were digitized at 10–20 kHz and filtered at 2.6 kHz. The pipette solution contains (in mM): K-gluconate 115; KCl 25; $CaCl_2$ 0.1; $MgCl_2$ 5; BAPTA 1; HEPES 10; $Na_2ATP$ 5; $Na_2GTP$ 0.5; cAMP 0.5; cGMP 0.5, pH7.2 with KOH,~320 mOsm. cAMP and cGMP were included to maintain the activity and longevity of the preparation. The bath solution consists of (in mM): NaCl 150; KCl 5; $CaCl_2$ 5; $MgCl_2$ 1; glucose 10; sucrose 5; HEPES 15, pH7.3 with NaOH,~330 mOsm. Chemicals and blockers were obtained from Sigma unless stated otherwise. Experiments were performed at room temperatures (20–22°C).

Optogenetic stimulation of transgenic animals was performed with an LED lamp, at 470 nm (from 8 mW/mm$^2$) for *hpIs166* and *hpIs279*, and at 625 nm (from 1.1 mW/mm$^2$), for *hpIs569* and *hpIs578*, respectively, controlled by the Axon amplifier software. One-second light exposure, a condition established by our previous study (*Gao et al., 2015*), was used to evoke the PSC bursts. The frequency power spectrum of rPSC bursts was analyzed using Clampfit 10.

## Statistical analysis

The Mann-Whitney U test, two-tailed Student's *t* test, one-way ANOVA test, or the Kolmogorov-Smirnov test were used to compare data sets and specified in figure legends. $p < 0.05$ was considered to be statistically significant; *, ** and *** denote $0.01 < p < 0.05$, $0.001 < p < 0.01$, $p < 0.001$, respectively. Graphing and subsequent analysis were performed using Igor Pro (WaveMetrics, Portland, OR), Clampfit (Molecular Devices), Image J (National Institutes of Health), R (http://www.R-project.org.), Matlab (MathWorks, Natick, MA), and Excel (Microsoft, Seattle, WA). For electrophysiology and calcium imaging, unless specified otherwise, each recording trace was obtained from a different animal; data were presented as the mean ± SEM.

## Acknowledgements

We thank Y. Wang, A. Liu, S. Teng, and J. R. Mark for technical assistance, the *Caenorhabditis Genetics Center* and National Bioresource Project for strains, C. Bargmann for the UNC-2 cDNA clone, S. Takayanagi-Kiya for *juIs440*. We thank Q. Wen, A. Samuel and A. Chisholm for discussions and comments on the manuscript. This work was supported by The National Natural Science Foundation of China (NSFC 31671052), Wuhan Morning Light Plan of Youth Science and Technology (2017050304010295) and the Junior Thousand Talents Program of China (S Gao), the National Institute of Health (C F-Y, MA, YJ, MZ), and the Canadian Institute of Health Research and the Natural Sciences and Engineering Research Council of Canada (MZ).

## Additional information

### Funding

| Funder | Grant reference number | Author |
| --- | --- | --- |
| Canadian Institutes of Health Research | MOP93619 MOP123250 | Mei Zhen |
| Natural Sciences and Engineering Research Council of Canada | 262112-12 | Mei Zhen |
| National Institutes of Health | R01-NS-082525 | Mei Zhen |
| National Natural Science Foundation of China | 31671052 | Shangbang Gao |
| Junior Thousand Talents Program of China | 0222170004 | Shangbang Gao |
| Wuhan Morning Light Plan of Youth Science and Technology | 2017050304010295 | Shangbang Gao |

| National Institutes of Health | GM084491 | Mark Alkema |
| National Institutes of Health | R01-NS-035546 | Yishi Jin |
| National Institutes of Health | R01-NS-084835 | Christopher Fang-Yen |

The funders had no role in study design, data collection and interpretation, or the decision to submit the work for publication.

## Author contributions

Shangbang Gao, Formal analysis, Supervision, Funding acquisition, Investigation, Methodology, Writing—original draft, Project administration, Writing—review and editing; Sihui Asuka Guan, Wesley Hung, Formal analysis, Investigation, Methodology, Writing—review and editing; Anthony D Fouad, Formal analysis, Investigation, Writing—original draft, Writing—review and editing; Jun Meng, Formal analysis, Validation, Investigation, Writing—review and editing; Taizo Kawano, Software, Methodology, Writing—review and editing; Yung-Chi Huang, Yi Li, Investigation; Salvador Alcaire, Formal analysis, Investigation; Yangning Lu, Investigation, Methodology; Yingchuan Billy Qi, Resources, Writing—review and editing; Yishi Jin, Methodology, Writing—review and editing; Mark Alkema, Supervision, Funding acquisition, Writing—review and editing; Christopher Fang-Yen, Supervision, Funding acquisition, Investigation, Writing—review and editing; Mei Zhen, Conceptualization, Resources, Formal analysis, Supervision, Funding acquisition, Investigation, Methodology, Writing—original draft, Project administration, Writing—review and editing

## Author ORCIDs

Shangbang Gao http://orcid.org/0000-0001-5431-4628
Yingchuan Billy Qi http://orcid.org/0000-0002-4267-4770
Yishi Jin http://orcid.org/0000-0002-9371-9860
Christopher Fang-Yen http://orcid.org/0000-0002-4568-3218
Mei Zhen http://orcid.org/0000-0003-0086-9622

## Decision letter and Author response

Decision letter https://doi.org/10.7554/eLife.29915.032
Author response https://doi.org/10.7554/eLife.29915.033

## Additional files

### Supplementary files
• Transparent reporting form
DOI: https://doi.org/10.7554/eLife.29915.025

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
