## [Decision Letter]

Thank you for submitting your article "Excitatory Motor Neurons are Local Central Pattern Generators in an Anatomically Compressed Motor Circuit for Reverse Locomotion" for consideration by *eLife*. Your article has been favorably evaluated by Eve Marder (Senior Editor) and three reviewers, one of whom, Ronald L Calabrese (Reviewer #1), is a member of our Board of Reviewing Editors. The following individuals involved in review of your submission have agreed to reveal their identity: Akinao Nose (Reviewer #2); David Biron (Reviewer #3).

The reviewers have discussed the reviews with one another and the Reviewing Editor has drafted this decision to help you prepare a revised submission.

Summary:

This is a very interesting manuscript that is very important for the *C. elegans* model system, but has real general interest as well. It dissects the motor system of *C. elegans* by selective cell ablation and mutant analysis and uses behavior and electrophysiology as assays to identify and define the reverse locomotion "CPG". While CPG may not be the right term in the minds of many, the authors do indeed identify a source of endogenous oscillation in motor neurons that underlies rhythmic reverse locomotion. For reverse locomotion the conclusion must be that the A-MNs, which are known to be dedicated to reverse locomotion, form a chain of motor neuron oscillators. The implication is that the B-MNs motor neurons, which are known to be dedicated to forward locomotion, form a similar chain of oscillators, but it is not nailed down completely and a head oscillator certainly exists that is not characterized. The limitation of the detailed analysis to reverse locomotion is not a real drawback, however, because the analysis uncovers important properties of the reverse locomotion circuit. Membrane potential oscillations are supported by HVA Ca channels (UNC-2), and very interestingly, premotor interneurons are not necessary for reverse (or forward) locomotion but exert state dependent control over A-MN oscillations and reverse locomotion initiation and maintenance. A-type-INs make mixed heterotypic and rectifying gap junction and excitatory chemical synapses on A-MNs to exert this control. At rest, AVA-A gap junctions dampen the excitability of coupled premotor INs and oscillatory activity of MNs to reduce propensity for reversals. Upon activation, AVA potentiate AMNs predominantly through excitatory chemical synapses, with a minor contribution from gap junctions. This is a truly interesting and important finding and should be more front and center in Discussion.

The work is very carefully and thoroughly done with appropriate controls, and the data analyzed appropriately. Necessary data appear in figures in an easily accessible form.

Essential revisions:

1) Please explain why B-MNs, in addition to premotor INs, have to be ablated for the A-MN to show intrinsic rhythmic activity (Figure 4). If the authors' model (Figure 7) is correct, the presence or absence of B-MNs should have no consequence on the intrinsic activity of the A-MN.

2) Subsection “A-MNs exhibit oscillatory activity independent of premotor IN inputs”, last paragraph: The authors state that all A-MNs show oscillating activity but with differences in the intensity. However, the authors only show results on DA9 in the figure (Figure 4). Can other A-MNs be illustrated, perhaps in supplementary figures to this figure? We also wonder what the phase relations are among the A-MNs in different segments and between antagonistic A-MNs (e.g., VA10 and DA7). Are they activated simultaneously, with phase difference, or with no phasic relationship? This is an important point in considering the nature of the CPG.

3) The full distributions of velocities (e.g., Figure 2) are useful. It would be convenient to also have summary statistics available at a glance. Could mean velocities and respective standard deviations be added to each of the velocity distribution panels throughout the manuscript? Preferably, on the figure itself as opposed to the caption/text. This seems more useful than p<0.001, which is quite obvious from the plot.

4) Subsection “A-MNs exhibit oscillatory activity independent of premotor IN inputs”, fourth paragraph: the data in Figure 5—figure supplement 1 is interpreted without much justification. Why did peaks in A-20% qualify as rPSC bursts while peaks in B-1% didn't? What were the criteria?

5) Subsection “A-MNs exhibit oscillatory activity independent of premotor IN inputs”, last paragraph: calcium imaging was performed in L4 larvae as opposed to measuring rPSCs in adult preps. Was this due to body thickness? Could development have contributed to the difference in the frequencies? Is there a difference in reversal behavior between L4 and adult B-MN / INs ablated worms?

6) The Results section "UNC-2 is an endogenous constituent of A-MN oscillation" is not sufficiently clear regarding the calcium flows. The second paragraph states that loss of function of the VGCC UNC-2 severely reduces calcium oscillations. Prior to explaining Figure 6, it bears mentioning that UNC-2 is thought to play two separable roles in A-MNs: promoting synaptic transmission and somatic Ca oscillations. This notion is supported by localizing UNC-2 to MN somas but that is only mentioned at the end of the section. Similarly, adding "despite the somatic oscillations remaining intact" to the end of the first paragraph would clarify. Finally, could the authors comment about the origin of the calcium they observe? The key role of UNC-2 seems to suggest that it is predominantly extracellular. Is this a valid conclusion?

7) A simple prediction of the model is that in the absence of B-MNs/INs or if AVA is optogenetically activated, *unc-7* mutants should exhibit increased reversal speed and/or persistence. Unfortunately, Figure 7 does not contain *unc-7* behavioral (reversal velocity/persistence) data. The optogenetic experiment may well be too onerous. Scoring reversal speed/persistence for an existing strain or explaining why the simple prediction is wrong may fall within the scope of this work.

8) "[P]ersistent body bends in premotor IN-less animals mainly originated from A-MN activity." Figure 2 seems to suggest that B-less worms are quite persistent (albeit slow) in their forward motion. Are the authors referring to the higher curvature of the backward-motion body waves here? If so, it is not clear. Similarly, Figure 2 shows that ~25% of forward motion is replaced by pauses (as compared to reversals). Could this just be a result of the fact that the independent oscillator(s) in the head are somewhat effective at interfering with forward body waves while no equivalent independent oscillator resides in the tail?

9) The manuscript does not comment on the ~50s calcium oscillation cycle in A-MNs. Does it correspond to some behavioral timescale or have a hypothesized significance? Do the mean velocities seem related? It may be the case that there is not much to say on the subject.

10) The writing needs reconsideration. Much of the important information is in the figure legends and the paper is thus tedious to follow. Putting more description of the figure in the text would help. The figures are very complicated and the data could be separated out into smaller figures. Word choice is not always the best and led to confusion for this reviewer. Many detailed examples of words that need clarifying are made on the manuscript pdf in Discussion and Introduction. The kymograms could be better explained so that readers unfamiliar with them can understand what the complex patterns in these kymograms from ablated and mutant animals mean.

11) The concept of CPG is not well articulated and its meaning is stretched a bit. You are really dealing with individual neuronal oscillators and not CPGs and equating the structure of the reverse locomotion circuit in *C. elegans* with vertebrate spinal cord or even invertebrate networks like that for leech swimming will be a hard sell to a veteran CPG person like myself. To my mind this is not the real interest of the paper. There are two major impacts arising from this work. 1) You have solved a pressing problem in *C. elegans* neurobiology which has plagued the field for over a decade. This should be emphasized. 2) You have made a truly brilliant discovery of the state dependent regulation of A-MN oscillations by the conjoint electrical and chemical synapses of premotor interneurons. This should be a centerpiece of the paper. The ideas on circuit compression are not compelling. Is the circuit compressed in *C. elegans* or is it expanded by evolution in leech? The first neurons in evolution were probably multifunctional (sensory-motor) and then evolution led to proliferation of neurons and specialization. That invertebrate neurons are very multifunctional has to be a truism since I was a graduate student and that motor neurons can be oscillatory has also been known as long. These arguments should be deemphasized.

12) Title could be shortened and refocused to match the revised Discussion.

---

## [Author Response]

Summary:This is a very interesting manuscript that is very important for the C. elegans model system, but has real general interest as well. It dissects the motor system of C. elegans by selective cell ablation and mutant analysis and uses behavior and electrophysiology as assays to identify and define the reverse locomotion "CPG". While CPG may not be the right term in the minds of many, the authors do indeed identify a source of endogenous oscillation in motor neurons that underlies rhythmic reverse locomotion.

Thank you. We are pleased to make contributions to our understanding on how motor circuits operate. We agree that in most context of this study, oscillator is a more suitable choice. We have made these changes, and are grateful for the suggestion.

For reverse locomotion the conclusion must be that the A-MNs, which are known to be dedicated to reverse locomotion, form a chain of motor neuron oscillators. The implication is that the B-MNs motor neurons, which are known to be dedicated to forward locomotion, form a similar chain of oscillators, but it is not nailed down completely and a head oscillator certainly exists that is not characterized. The limitation of the detailed analysis to reverse locomotion is not a real drawback, however, because the analysis uncovers important properties of the reverse locomotion circuit.

Indeed. Although we chose to focus on oscillators for the reverse motor circuit, we could not address it without first knowing that different oscillators underlie forward and reverse movements (Figure 1 and Figure 2). This concept may sound trivial, but it was not established when we began the study.

Another critical piece of information from Figure 2 is that although both A- and B-class motor neurons may function as oscillators, they differ in their intrinsic physiological properties: A-class motor neurons exhibit high intrinsic oscillatory activities in the absence of premotor interneurons.

Such a distinction alludes to a potential difference in the functional relationships between the two classes of motor neurons with their respective input interneurons. More important is the notion that only when the A-MNs activity was removed could the intrinsic property of the B-MNs to be addressed in isolation (Figure 2). Establishing these conceptual frameworks allowed we and others to design mechanistic studies to probe either the forward or reverse motor circuit.

We had chosen to focus on the reverse motor circuit for two reasons: 1) examining the functional relationship between A motor neurons and the descending premotor interneurons has been a long-standing goal of our group; 2) we would like to collaborate with and help other groups who are interested in the *C. elegans* motor circuit.

Membrane potential oscillations are supported by HVA Ca channels (UNC-2), and very interestingly, premotor interneurons are not necessary for reverse (or forward) locomotion but exert state dependent control over A-MN oscillations and reverse locomotion initiation and maintenance. A-type-INs make mixed heterotypic and rectifying gap junction and excitatory chemical synapses on A-MNs to exert this control. At rest, AVA-A gap junctions dampen the excitability of coupled premotor INs and oscillatory activity of MNs to reduce propensity for reversals. Upon activation, AVA potentiate AMNs predominantly through excitatory chemical synapses, with a minor contribution from gap junctions. This is a truly interesting and important finding and should be more front and center in Discussion.

Thank you. We have revised the Abstract, Introduction, and Discussion to emphasize this finding. Further, we present additional calcium imaging and behavioural results that support the state-dependent dual-regulation model (Figure 9 and Figure 9—figure supplement 1 in the revised manuscript).

The work is very carefully and thoroughly done with appropriate controls, and the data analyzed appropriately. Necessary data appear in figures in an easily accessible form.Essential revisions:1) Please explain why B-MNs, in addition to premotor INs, have to be ablated for the A-MN to show intrinsic rhythmic activity (Figure 4). If the authors' model (Figure 7) is correct, the presence or absence of B-MNs should have no consequence on the intrinsic activity of the A-MN.

Sorry for the confusion. A-MN’s intrinsic activity does not depend on the presence or absence of B-MNs; B-MNs only interfere with their propagation after premotor INs are removed. In Figure 4 and others, the purpose of removing B-MNs along with premotor interneurons was to ensure that rPSC signals that we examined were from A-MNs.

The behavioural phenotype of premotor IN ablated animals (Kawano et al., 2011; this study), either alone or in combination with A-MNs or B-MNs (Figure 1; Figure 2) indicate that in the absence of premotor INs, A-MNs and B-MNs underlie the antagonizing driving force for reverse and forward movements, respectively. The shallow bending after the co-ablation of premotor interneurons and A-MNs indicates that B-MNs have lower intrinsic activity. A previous study showed that body bending can activate B-MNs (Wen et al., 2012). Hence, if A-MNs remain active in the absence of premotor interneurons, the resulted body bending may subsequently activate B-MNs that reside at the region. Without removing B-MNs, we could not be certain that the observed oscillatory signals were from A-MNs.

Consistent with this notion, we performed the complementary experiment. In preparations where we removed A-MNs along with premotor INs, the mini frequencies were drastically reduced, but the rPSC burst was not increased (Author response image 1). These results confirmed that rPSCs we observed upon removing premotor INs were primarily driven by A-MNs. We explained this point (subsection “A-MNs exhibit oscillatory activities independent of premotor IN inputs”, third paragraph).

**Author response image 1. respfig1:** Co-ablation of premotor INs and A-MNs did not increase the rPSC burst. (**A**) Example traces of postsynaptic currents (recorded at -60 mV) in animals without (Control) and with (Ablated) premotor IN- and A-MN-ablation.(B, C) Quantification of the PSC burst and minature PSC (mPSC) frequency, without (Ctrl) and with (Ablated) the ablation of premotor INs and A-MNs. PSC events were rare in both preparations, while the mPSC frequency was drastically reduced. n = 10 animals (Control), n = 11 animals (Ablated). * P < 0.05 against Control by the Mann-Whitney U test. Error bars, SEM.

2) Subsection “A-MNs exhibit oscillatory activity independent of premotor IN inputs”, last paragraph: The authors state that all A-MNs show oscillating activity but with differences in the intensity. However, the authors only show results on DA9 in the figure (Figure 4). Can other A-MNs be illustrated, perhaps in supplementary figures to this figure? We also wonder what the phase relations are among the A-MNs in different segments and between antagonistic A-MNs (e.g., VA10 and DA7). Are they activated simultaneously, with phase difference, or with no phasic relationship? This is an important point in considering the nature of the CPG.

In Figure 4, the DA9 calcium imaging experiments were performed in animals immobilized by glue. This preparation was to examine A-MN activity with minimal proprioceptive feedback or coupling. DA9 exhibited the most robust calcium signals than other A-MNs (Author response image 2).

**Author response image 2. respfig2:** An example trace of simultaneous DA9 and VA11 calcium oscillation in an immobolized animal after the ablation of premotor INs/B-MNs.

We thought that the phasic relationships among multiple A-MNs should be analyzed in moving animals. This was why we performed simultaneous calcium imaging data for three A-MNs (VA10, DA7 and VA11) in animals where we removed all premotor INs and B-MNs, and when animals were moving (Figure 2 in the original manuscript, Figure 5 in the revised manuscript). We show that these A-MNs exhibited phasic relationships that correlated with alternative dorsal/ventral bending and anterior/posterior propagation during movement. Specifically, DA7, which activates dorsal muscles, exhibited activity changes that were anti-phasic to that of ventral muscle activators (VA10 and VA11), whereas VA11’s activity changes preceded that of VA10.

To clearly convey the information, we now show these results in a separate figure, with an illustration of the spatial relationship of these neurons(Figure 5 in the revised manuscript).

3) The full distributions of velocities (e.g., Figure 2) are useful. It would be convenient to also have summary statistics available at a glance. Could mean velocities and respective standard deviations be added to each of the velocity distribution panels throughout the manuscript? Preferably, on the figure itself as opposed to the caption/text. This seems more useful than p<0.001, which is quite obvious from the plot.

Thank you for this helpful suggestion. We added mean velocities and deviations in figures where they help to clarify their effect on animals’ motility (Figure 1) or velocity (Figure 8). The key message for other figures such as Figure 2 was the change in preference of direction: in the absence of premotor interneurons, ablation of B-MNs (Figure 2) or A-MNs (Figure 2ii) led to predominantly forward or reversal movements, whereas ablating D-MNs (Figure 2iii) did not change the ‘kink’ state. So we did not show mean velocities in these figures.

4) Subsection “A-MNs exhibit oscillatory activity independent of premotor IN inputs”, fourth paragraph: the data in Figure 5—figure supplement 1 is interpreted without much justification. Why did peaks in A-20% qualify as rPSC bursts while peaks in B-1% didn't? What were the criteria?

Sorry for the confusion. We applied the criteria that were defined in a previous study (Gao et al., 2015). Briefly, rPSC bursts consisted of a train of depolarization events (-50 – -300 pA, 3–10 Hz), which is distinct from the high-frequency monophasic and multiphasic mPSC events (subsection “A-MNs exhibit oscillatory activities independent of premotor IN inputs”, second paragraph). Events in Figure 5—figure supplement 1 (B-1%) fell in the second category (Author response image 3).

**Author response image 3. respfig3:** Power spectrum analysis of the current traces in Figure 5—figure supplement 1 A-20% and B-1% revealed no ~10Hz rPSCs burst phase in B-1%.

5) Subsection “A-MNs exhibit oscillatory activity independent of premotor IN inputs”, last paragraph: calcium imaging was performed in L4 larvae as opposed to measuring rPSCs in adult preps. Was this due to body thickness? Could development have contributed to the difference in the frequencies? Is there a difference in reversal behavior between L4 and adult B-MN / INs ablated worms?

We used L4 animals for calcium imaging analyses for two reasons: 1) we used L4 animals for behavioural analyses; 2) the Punc-4 promoter that we used to express the calcium sensor is strong in A-MNs in L4, but falls off sharply afterwards, making recording in adults more noisy. We don’t think that the developmental difference leads to the two-fold difference of the frequency of rPSC and calcium oscillation. In a few DA9 calcium imaging recordings that we obtained in young adults (Author response image 4), they exhibited similar frequency as in L4.

**Author response image 4. respfig4:** Example traces of DA9 calcium oscillation in a young adult before (Control) and after ablation (Ablated) of premotor INs/B-MNs.

6) The Results section "UNC-2 is an endogenous constituent of A-MN oscillation" is not sufficiently clear regarding the calcium flows. The second paragraph states that loss of function of the VGCC UNC-2 severely reduces calcium oscillations. Prior to explaining Figure 6, it bears mentioning that UNC-2 is thought to play two separable roles in A-MNs: promoting synaptic transmission and somatic Ca oscillations. This notion is supported by localizing UNC-2 to MN somas but that is only mentioned at the end of the section. Similarly, adding "despite the somatic oscillations remaining intact" to the end of the first paragraph would clarify.

Thank you. We have revised this section accordingly.

Finally, could the authors comment about the origin of the calcium they observe? The key role of UNC-2 seems to suggest that it is predominantly extracellular. Is this a valid conclusion?

These results are indeed consistent with UNC-2 being the origin of extracellular calcium. We did not make an explicit claim previously because of the possibility that release from the intracellular calcium store, upon such an extracellular entry, also contribute to the observed calcium profile in DA9 soma. We have now clarified this point (e.g. subsection “The P/Q/N-type VGCC underlies A-MN oscillation independently of synaptic transmission”, second paragraph).

7) A simple prediction of the model is that in the absence of B-MNs/INs or if AVA is optogenetically activated, unc-7 mutants should exhibit increased reversal speed and/or persistence. Unfortunately, Figure 7 does not contain unc-7 behavioral (reversal velocity/persistence) data. The optogenetic experiment may well be too onerous. Scoring reversal speed/persistence for an existing strain or explaining why the simple prediction is wrong may fall within the scope of this work.

Thank you for these suggestions. We performed new experiments on *unc-7* mutants, results from which further validate the hypothesis that AVA premotor INs inhibit A-MN activity through gap junctions to reduce propensity for the reverse motor state.

First, AVA-specific restoration of UNC-7 was sufficient to restore inhibition of A-MN oscillation in *unc-7* mutants (Figure 9 in revised manuscript). Previously, we showed that in *unc-7* mutants, DA9 exhibited calcium oscillation in the presence of premotor INs, implicating that AVA failed to inhibit A-MN’s activity in the absence of gap junction coupling. Because UNC-7 expression is not restricted to AVA, we performed the rescue experiment to confirm that this effect was due to the specific loss of UNC-7 in AVA. Through the FLT-FRT system, we generated AVA-specific restoration of UNC-7 in *unc-7* mutants, and found that DA9 calcium oscillation was indeed attenuated (Figure 9 in revised manuscript).

Next, we performed behavioural analyses to examine the effect of AVA-A gap junction coupling. In the presence of premotor INs/B-MNs (Figure 9 in revised manuscript), as predicted from the model, *unc-7* mutants exhibited a drastic increase in the propensity for reverses at the expense of forward movement; the restoration of UNC-7 in AVA alone restored the animal’s preference for the forward movement.

8) "[P]ersistent body bends in premotor IN-less animals mainly originated from A-MN activity." Figure 2 seems to suggest that B-less worms are quite persistent (albeit slow) in their forward motion. Are the authors referring to the higher curvature of the backward-motion body waves here? If so, it is not clear.

Sorry about the lack of clarity. We reworded relevant sections in the revised manuscript; a more detailed explanation follows.

In the absence of premotor INs, animals continued to exhibit body bends (Figure 1). When we further ablated A-MNs, animals were able to move forward slowly, but with little bending (Figure 2Bii). When we further ablated B-MNs, premotor INs-less animals executed reverse movement with body bending. These results suggest that A-MNs have higher intrinsic activities than B-MNs.

But in the absence of premotor interneurons, B-MNs strongly antagonize A-MN-driven reverse movements. Because B-MNs are activated by body bending (Wen et al., 2012), a simple interpretation is that in the absence of premotor INs, body bending primarily results from the high intrinsic A-MNs, which may subsequently potentiate B-MNs that reside in the body segment to interfere with their propagation.

Similarly, Figure 2 shows that ~25% of forward motion is replaced by pauses (as compared to reversals). Could this just be a result of the fact that the independent oscillator(s) in the head are somewhat effective at interfering with forward body waves while no equivalent independent oscillator resides in the tail?

While the ‘pause’ (Figure 2) describes the lack of centroid displacement in either A- or B-ablated animals, causes for pauses were different. We consistently observed that 1) premotor IN/A-MN ablated animals moved forward slowly and stalled frequently; 2) premotor IN/B-MN ablated animals reversed with deep body bends, but their reversals were often interrupted by head bends that tried to pull the anterior body forward, and the pull hindered mid-point displacement (Video 2).

We agree with the reviewer that these behaviour results suggest that there is no equivalent independent reversal oscillator in the tail. Consistent with this notion, animals without premotor INs or MNs could exhibit high frequency oscillation of the head, but not of the tail (Figure 1).

We refrained from elaborating on these hypotheses because they suggest complex, yet-to-be addressed relationships between the forward and reversal oscillators. We hope to resolve it in future studies.

9) The manuscript does not comment on the ~50s calcium oscillation cycle in A-MNs. Does it correspond to some behavioral timescale or have a hypothesized significance? Do the mean velocities seem related? It may be the case that there is not much to say on the subject.

In our calcium imaging experiments on crawling animals, after the ablation of premotor INs/B-MNs, they exhibited ~10s reversal undulation cycle on plates (Figure 5 in revised manuscript). DA9’s ~50s oscillatory cycle was observed in glued animals; the 5-fold difference in frequency likely reflects the effect of proprioceptive feedback.

Despite the difference in their absolute value, A-MN’s calcium oscillation is positively correlated with the animal’s undulation frequency, hence velocity in crawling animals (Figure 5 in revised manuscript). The loss and gain-of-function unc-2 mutants exhibited lengthened and shortened DA9’s oscillation, respectively (Figure 7 in revised manuscript); they exhibited decreased and increased reversal undulation frequency and velocity (Figure 8 in revised manuscript). The difference between DA9 calcium oscillation and undulation frequency was also ~5 fold for unc-2(gf) mutants. We commented on the correlation between A-MN calcium oscillation and velocity (e.g. subsection “A-MNs exhibit oscillatory activities independent of premotor IN inputs”, last paragraph; subsection “A-MNs are rhythm generators”, first paragraph).

10) The writing needs reconsideration. Much of the important information is in the figure legends and the paper is thus tedious to follow. Putting more description of the figure in the text would help. The figures are very complicated and the data could be separated out into smaller figures. Word choice is not always the best and led to confusion for this reviewer. Many detailed examples of words that need clarifying are made on the manuscript pdf in Discussion and Introduction. The kymographs could be better explained so that readers unfamiliar with them can understand what the complex patterns in these kymographs from ablated and mutant animals mean.

Thank you for the careful, constructive editorial comments. We thrive to make the description and illustration of ours and other’s work precise and clear. We have incorporated suggested editions, provided more descriptions for behavioural analyses, and reorganized figures to convey key messages.

11) The concept of CPG is not well articulated and its meaning is stretched a bit. You are really dealing with individual neuronal oscillators and not CPGs and equating the structure of the reverse locomotion circuit in C. elegans with vertebrate spinal cord or even invertebrate networks like that for leech swimming will be a hard sell to a veteran CPG person like myself. To my mind this is not the real interest of the paper.

We appreciate these comments. In the previous manuscript, we indeed did not make sufficient distinction between the usage of ‘CPG’ and ‘oscillator’. This in part reflects where the field was: the presence of oscillators was debated. Establishing that *C. elegans* motor rhythm is CPG-driven is important for *C. elegans* motor circuit, but we agree with you that ‘oscillator’ is a more appropriate word to define A-MNs’ activity.

There are two major impacts arising from this work. 1) You have solved a pressing problem in C. elegans neurobiology which has plagued the field for over a decade. This should be emphasized. 2) You have made a truly brilliant discovery of the state dependent regulation of A-MN oscillations by the conjoint electrical and chemical synapses of premotor interneurons. This should be a centerpiece of the paper. The ideas on circuit compression are not compelling. Is the circuit compressed in C. elegans or is it expanded by evolution in leech? The first neurons in evolution were probably multifunctional (sensory-motor) and then evolution led to proliferation of neurons and specialization. That invertebrate neurons are very multifunctional has to be a truism since I was a graduate student and that motor neurons can be oscillatory has also been known as long. These arguments should be deemphasized.

Thank you for the compliment to our work. We have revised the title, Abstract, Introduction, figures, and Discussion to emphasize and elaborate on the regulatory mechanisms on A-MN’s oscillatory activity. We further highlighted the similarity between our findings and previous discoveries in other small motor circuits.

We understand and respect your reservation with the usage of the word ‘compression’. We did not mean to use it in the context of evolution, but wanted to describe a nervous system property that we feel to be more specific than ‘multifunctionality’ (e.g. Getting, 1989). To our understanding, multifunctionality could also be used to describe the same group of neurons or connections to be reconfigured or recruited to generate different motor programs in the same animal. When *C. elegans* excitatory motor neurons play not only the roles of inducing muscle contraction, but also rhythm-generation, and possibly proprioceptive feedback and coupling, we feel that ‘compression’ better emphasizes the reduction of circuit layers by packing their functions into a single class of neurons.

12) Title could be shortened and refocused to match the revised Discussion.

The title has been shortened.